



**Shell chemistry of the Boreal Campanian bivalve *Rastellum diluvianum* (Linnaeus,**
**1767) reveals temperature seasonality, growth rates and life cycle of an extinct**
**Cretaceous oyster.**
Niels J. de Winter[1], Clemens V. Ullmann[2], Anne M. Sørensen[3], Nicolas R. Thibault[4], Steven
Goderis[1], Stijn J.M. Van Malderen[5], Christophe Snoeck[1,6], Stijn Goolaerts[7], Frank Vanhaecke[5],
Philippe Claeys[1]
[1]AMGC research group, Vrije Universiteit Brussel, Pleinlaan 2, B-1050 Brussels, Belgium.
[2]Camborne School of Mines, University of Exeter, Penryn, Cornwall, TR10 9FE, UK.
[3]Trap Danmark, Agem All 13, DK-2970, Hørsholm, Denmark.
[4] Department of Geoscience and Natural Resource Management, University of Copenhagen,
Øster voldgade 10, DK-1350 Copenhagen C., Denmark.
[5]A&MS research unit, Ghent University Campus Sterre, Krijgslaan 281, Building S12, B-9000
Ghent, Belgium.
[6]G-Time laboratory, Université Libre de Bruxelles, 50 Avenue F.D. Roosevelt, B-1050, Brussels,
Belgium.
[7]Directorate of Earth and History of Life, Royal Belgian Institute of Natural Sciences, Vautierstraat
29, B-1000 Brussels, Belgium.
*Correspondence to: Niels J. de Winter (niels.de.winter@vub.be)*



**Abstract**

The Campanian age (Late Cretaceous) is characterized by a warm greenhouse climate with limited land
ice volume. This makes the Campanian an ideal target for the study of climate dynamics during greenhouse
periods, which are essential for predictions of future climate change due to anthropogenic greenhouse gas
emissions. Well-preserved fossil shells from the Campanian age (±78 Ma) high paleolatitude (50°N) coastal
faunas of the Kristianstad Basin (southern Sweden) offer unique snapshot of short-term climate and
environmental variability during the Campanian, which complement traditional long-term climate
reconstructions. In this study, we apply a combination of high-resolution spatially resolved trace element
analyses (μXRF and LA-ICP-MS), stable isotope analyses (IRMS) and growth modelling to study short-
term (seasonal) variations recorded in the oyster species *Rastellum diluvianum* from Ivö Klack. A
combination of trace element and stable isotope records of 12 specimens sheds light on the influence of
specimen-specific and age-specific effects on the expression of seasonal variations in shell chemistry and
allows disentangling vital effects from environmental influences in an effort to refine palaeoseasonality
reconstructions of Late Cretaceous greenhouse climates. Growth modelling based on stable isotope
records from *R. diluvianum* further allows to discuss the mode of life, circadian rhythm and reproductive
cycle of extinct oysters and sheds light on their ecology. This multi-proxy study reveals that mean annual
temperatures in the Campanian high-latitudes were 17 to 19°C with a maximum extent of seasonality of
14°C. These results show that the latitudinal gradient in mean annual temperatures during the Late
Cretaceous was steeper than expected based on climate models and that the difference in seasonal
temperature variability between latitudes was much smaller in the Campanian compared to today. Our
results also demonstrate that species-specific differences and uncertainties in the composition of Late
Cretaceous seawater prevent trace element proxies (Mg/Ca, Sr/Ca, Mg/Li and Sr/Li) to be used as reliable
temperature proxies for fossil oyster shells.

**1. Introduction**


The Late Cretaceous was marked by a long cooling trend that brought global mean annual temperatures
(MAT) down from the mid-Cretaceous climate maximum (±28°C surface ocean temperatures) in the
Cenomanian and Turonian (±95 Ma) to slightly cooler temperatures (±22°C surface ocean temperatures)
around the Campanian-Maastrichtian boundary (±72.1 Ma; Clarke and Jenkyns, 1999; Pearson et al., 2001;
Huber et al., 2002; Friedrich et al., 2012; Scotese, 2016). This cooling trend was likely caused by a change
in ocean circulation, initiated by the opening of the Equatorial Atlantic Gateway that separated the proto-
North and -South Atlantic Ocean basins (Friedrich et al., 2009). It is well recorded in the white chalk
successions of the Chalk Sea, which covered large portions of northwestern Europe during the Late
Cretaceous Period (Reid, 1973; Jenkyns et al., 1994; Jarvis et al., 2002; Voigt et al., 2010). These chalk
successions featured in various paleoclimate studies, because they are accessible in good outcrops and
consist predominantly of calcareous nannofossils which faithfully record sea surface conditions (e.g.
Jenkyns et al., 1994). Furthermore, the connection of the Chalk Sea to the (proto-)North Atlantic Ocean
makes it an interesting area of study to constrain Late Cretaceous paleogeography and climate. Even with
this prolonged cooling trend in the Late Cretaceous, proxy data and climate models show that the
Campanian was still characterized by a relatively warm global climate with a shallow equatorial temperature
gradient compared to today (Huber et al., 1995; Brady et al., 1998; Huber et al., 2002). Even though sea
level changes seem to indicate possible small changes in land ice volume during the Late Cretaceous,
warm high-latitude paleotemperatures seem to rule out the possibility of extensive polar ice sheets
comparable in volume to modern ice caps (Barrera and Johnson, 1999; Huber et al., 2002; Jenkyns et al.,
2004; Miller et al., 2005; Thibault et al., 2016). Given these climatic conditions and a relatively modern
continental configuration, the Campanian serves as an interesting analogue for Earth's climate in the near
future, should anthropogenic and natural emissions continue to contribute to the rise in global temperatures
and decrease global ice volume on Earth (IPCC, 2013; Donnadieu et al., 2016). Most Late Cretaceous
climate reconstructions focus on reconstructing and modelling long-term evolutions of humid/arid conditions
on land and/or past atmospheric and oceanic temperatures (DeConto et al., 1999; Thibault et al., 2016;
Yang et al., 2018). Data on the extent of seasonal variability from this time period, especially from high-
latitudes, are scarce, although such data constitute a fundamental component of the climate system
(Steuber, 1999; Steuber et al., 2005; Burgener et al., 2018).



Fossil bivalve shells offer a valuable record for studying past climates on a seasonal scale. The chemistry
of their shells records information on the environment in which bivalves grew, and incremental
measurements of chemical changes along the growth direction (sclerochronological studies) potentially
yield records of seasonal environmental changes (Mook, 1971; Jones, 1983; Klein et al., 1996a; Schöne
and Gillikin, 2013). Their distribution allows paleoseasonality reconstructions across a wide range of
latitudes (Roy et al., 2000; Jablonski et al., 2017), and the preservation potential of calcitic shell structures
(especially in oyster shells) makes them ideal, if not one of the only, recorders of pre-Quarternary
seasonality and sub-annual environmental change (Brand and Veizer, 1980; 1981; Al-Aasm and Veizer,
1986a; b; Immenhauser et al., 2005; Alberti et al., 2017). The incremental growth of bivalve shells in practice
means that the limits in terms of time resolution of reconstructions from bivalve shells are governed by
sampling resolution rather than the resolution of the record itself. While periods of growth cessation can
occur (especially in high latitudes, Ullmann et al., 2010), in practice this allows reconstructions of changes
down to sub-daily timescales given the right sampling techniques (Schöne et al., 2005; Sano et al., 2012;
Warter et al., 2018; de Winter et al., in review). Examples of chemical proxies used for these
paleoseasonality reconstructions include stable carbon and oxygen isotope ratios and trace element ratios
(e.g. Steuber et al., 2005; Gillikin et al., 2006; McConnaughey and Gillikin, 2008; Schöne et al., 2011; de
Winter et al., 2017a; 2018).
The incorporation of these chemical proxies into bivalve shells is challenged by the influence of so-called
vital effects: biological controls on the incorporation of elements in the shell independent of the environment
(Weiner and Dove, 2003; Gillikin et al., 2005). These vital effects have been shown to mask the
characteristic relationships between shell chemistry and the environment, and appear to be distinct not only
between different bivalve species but also between specimens of different ontogenetic age (Freitas et al.,
2008). Differences between bivalve families mean that the chemistry of some taxa (like scallops: Family
Pectinidae) are especially affected by vital effects (Lorrain et al., 2005; Freitas et al., 2008), while other
families like oysters (Family Ostreidae) seem to be more robust recorders of environmental conditions
(Surge et al., 2001; Surge and Lohmann, 2008; Ullmann et al., 2010; 2013). Nevertheless, the effect of
changes in microstructure and the amount of organic matrix present in different parts of (oyster) shells on
shell chemistry and preservation introduces uncertainty as to which parts of the shells are well-suited for
reconstruction purposes (Carriker et al., 1991; Kawaguchi et al., 1993; Dalbeck et al., 2006; Schöne et al.,
2010; 2013). The key to disentangling these vital effects from recorded environmental changes lies in the
application of multiple proxies and techniques on the same bivalve shells (the "multi-proxy approach"; e.g.
Ullmann et al., 2013; de Winter et al., 2017a; 2018) and to base reconstructions on more than one shell
(Ivany, 2012).
The Kristianstad Basin is located on the southeastern Baltic Sea coast of the southern Swedish province
of Skåne (56°2' N, 14° 9' E; see **Fig. 1**). Shallow marine sediments deposited at Ivö Klack consist of sandy
and silty nearshore deposits containing carbonate gravel and are coarsely dated in the latest early
Campanian (Christensen, 1975; 1984; Surlyk and Sørensen, 2010; Sørensen et al., 2015). The sediments
were deposited in a near-shore setting described as a rocky coastline that was inundated during the
maximum extent of the Late Cretaceous transgression, the paleolatitude is 50°N (Kominz et al., 2008; Csiki-
Sava et al., 2015). Since the region has remained tectonically quiet since the Campanian, the deposits of
Kristianstad Basin localities remain at roughly the same altitude as when they were deposited and have
been subject to limited burial (Surlyk and Sørensen, 2010). The rocky shore deposits of Ivö Klack are
characterized by a diverse shelly fauna, consisting of well-preserved fossils and fragments of brachiopods,
belemnites, echinoids and asteroids, polychaete worms, gastropods, corals, ammonites and thick-shelled
oysters, with a total of almost 200 different recognized species (Surlyk and Sørensen, 2010). In this diverse
rocky shore ecosystem, various habitat zones can be distinguished, each with their distinct suite of
organisms adapted to local conditions of varying amounts of sunlight, sedimentation and turbulence (Surlyk
and Christensen, 1974; Sørensen et al., 2012). This unique combination of marine biodiversity and
preservation of original shell material makes the localities in Kristianstad Basin ideal for studying sub-annual
variability in shell chemistry and reconstructing paleoseasonality and environmental change in the
Campanian (Sørensen et al., 2015).
In this study, we present a detailed, multi-proxy comparison of the growth and chemistry of well-preserved
fossil shells of the thick-shelled oyster *Rastellum diluvianum* (Linnaeus, 1767) recovered from the Ivö Klack
locality on the northern edge of the Kristianstad Basin. We combine stable isotope proxies conventional in





sclerochronological studies ($\delta^{13}$C and $\delta^{18}$O; e.g. Goodwin et al., 2001; Steuber et al., 2005) with less well-
established trace element proxies (Mg/Ca, Sr/Ca, Mg/Li and Sr/Li; e.g. Bryan and Marchitto, 2008; Schöne
et al., 2011; Füllenbach et al., 2015; Dellinger et al., 2018) and growth modelling based on $\delta^{18}$O seasonality
(Judd et al., 2018) in an attempt to disentangle the effects of growth rate, reproductive cycle and
environmental change on shell chemistry. The data gathered in this study allow a detailed discussion on
seasonal changes in temperature and water chemistry in the coastal waters of the Kristianstad Basin in the
late early Campanian, as well as on the life cycle of *R. diluvianum* and its response to seasonal changes in
its environment.

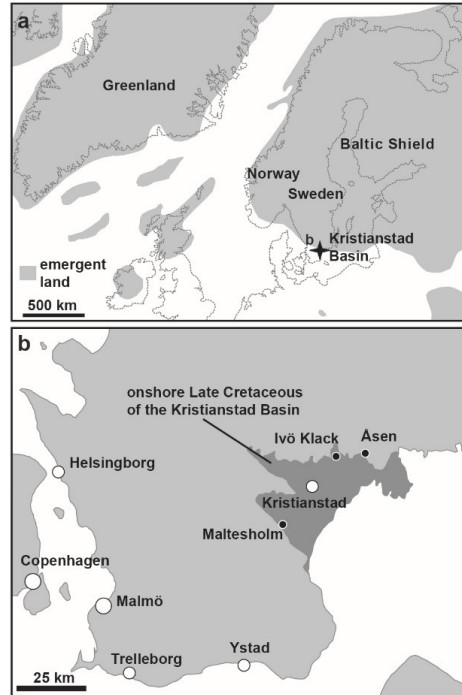

**Figure 1:** Paleogeographic map of the Boreal Chalk Sea (**a**) and the area of present-day southern Sweden (**b**) showing the location of Ivö klack (modified after Sørensen et al., 2015)

**2. Materials and Methods**
2.1 Sample acquisition and preparation
Complete valves of twelve individual *R. diluvianum* oysters were obtained from the Ivö Klack locality (see
**Fig. 2**). Specimens of *R. diluvianum* were found *in situ* attached to the vertical sides of large boulders that
characterized the rocky shore of Ivö Klack (Surlyk and Christensen, 1974). The valves were cleaned and
fully embedded in Araldite® 2020 epoxy resin (Bodo Möller Chemie Benelux, Antwerp, Belgium).
Dorsoventral slabs (±10 mm thick) were cut perpendicular to the hinge line using a water-cooled slow
rotating saw with a diamond-coated blade (thickness ± 1 mm; **Fig. 2**). The surfaces cut on the central growth
axis were progressively polished using silicon-carbide polishing disks (up to P2500, or 8.4 μm grain size).
Polished surfaces were scanned at high (6400 dpi) resolution using an Epson Perfection 1650 flatbed color
scanner (Seiko Epson Corp., Suwa, Japan). Resulting color scans of all polished *R. diluvianum* shell cross
sections are provided in **Fig. 2** and **S1**. Shell microstructures in *R. diluvianum* shells were studied in detail
on high-resolution scans and by using reflected light microscopy. Microstructural features were used to
reconstruct the relative timing of shell growth (see **Fig. 3**). Fragments of visually well-preserved material
from different microstructures in the shells were coated with gold and studied under a Scanning Electron
Microscope (Quanta 200 ESEM) and imaged at 1000x – 2000x magnification (**Fig. 3b-e**). Chemical



analyses were carried out on polished cross sections in order of sample size and destructive character of
sampling (starting with the least destructive measurements).
**Figure 2:** Overview of the 12 *Rastellum diluvianum* shells used in this study. All shells are depicted on the same scale (see scalebar
in center of image). Colors of the lines under sample names correspond to the colors of the lines in **Fig. 7** and **Fig. 9**. Every shell is
represented by an image of the inside of the valve analyzed, as well as a color scan of the cross section through the shell on which
high-resolution analyses were carried out. The dashed red line shows the location of these cross sections. The largest 5 shells (1-5,
on top half) were sampled for IRMS analyses ($\delta^{13}C$ and $\delta^{18}O$). All shells were subjected to micro X-ray fluorescence (µXRF), laser
ablation inductively coupled plasma mass spectrometry (LA-ICP-MS) and multi-cup inductively coupled plasma mass spectrometry
(MC-ICP-MS) analyses. Full-size versions of the high-resolution color scans of shell cross sections are provided in **S1**.

## Overview of *Rastellum diluvianum* shells

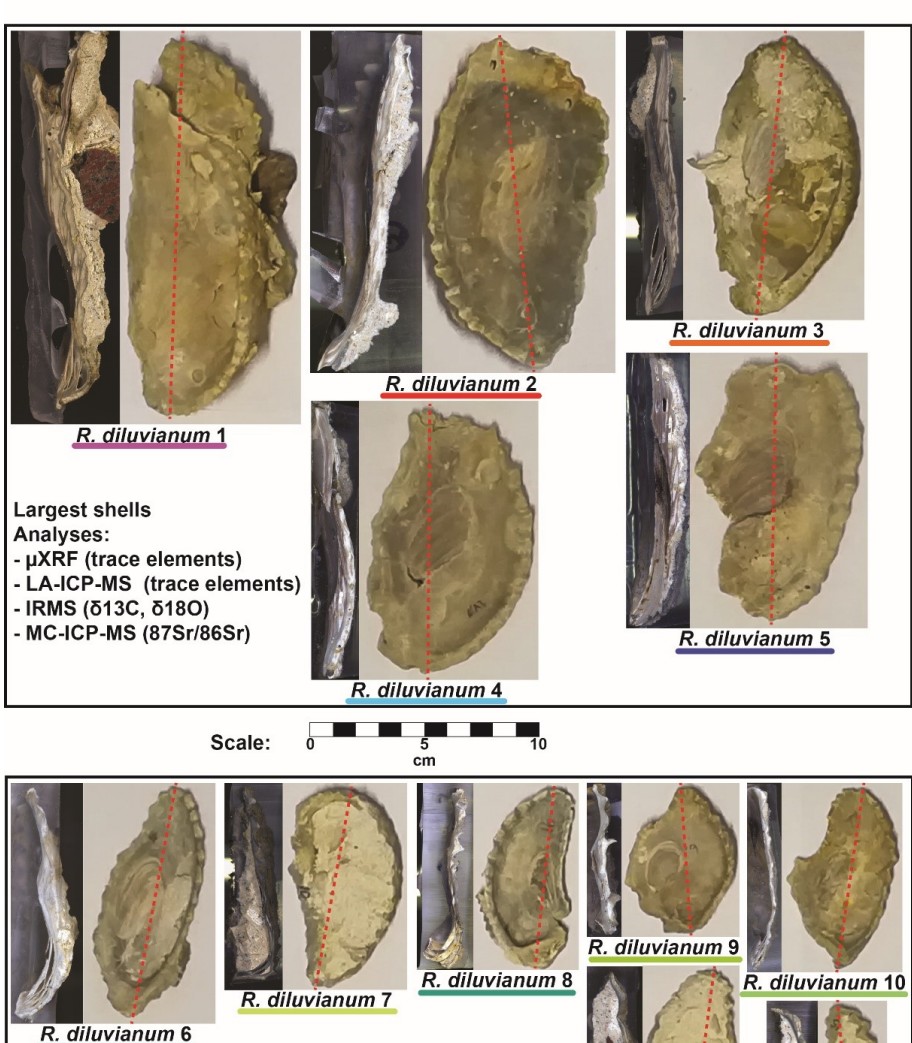

## *R. diluvianum* 3

[Figure 3 images a–h]

**Figure 3:** Overview image showing a high-resolution color scan of the cross section through *R. diluvianum* 3 (**a**) on which the different shell textures as well as the directions of high-resolution analyses (in growth direction) are indicated. Thin blue lines denote parts of the shell that were deposited at the same time (growth increments). (**b**) and (**c**) show SEM images of the well-preserved foliated calcite in the shell. More porous structures in the shell (vesicular calcite) are depicted in SEM images shown in (**d**) and (**e**). Below are shown three XRF elemental maps of the same cross section: A RGB-colored map displaying the relative abundances of Fe, Si and Ca (**f**), A heatmap of Fe concentrations (**g**; see scalebar below map) and a heatmap of Sr concentrations (**h**; see scalebar below map). XRF mapping only yields relative (semi-quantitative) abundance of elements.

## 2.2 Micro-XRF mapping

Elemental abundance maps of all *R. diluvianum* shell cross sections were obtained using a Bruker Tornado M4 energy-dispersive micro-X-Ray Fluorescence scanner (µXRF; Bruker nano GmbH, Berlin, Germany) All µXRF analyses carried out with the Bruker M4 Tornado are non-destructive. The µXRF is equipped with a Rh filament metal-ceramic tube X-Ray source operated at 50 kV and 600 µA (30 W; maximum energy



settings). The circular spot projected on the same surface is estimated to have a diameter of 25 μm (Mo-
Kα). A μm-precision XYZ translation stage allows for quick and precise sample movement such that a grid
of 25 μm XRF spots can be measured on the sample surface by continuous scanning to construct elemental
maps ($3 * 10^6$ - $5 * 10^6$ pixels per map). Exposure times of the X-ray beam per sampling position in mapping
mode (1 ms/pixel) are too short to gain adequate signal-to-noise ratio for pixel-by-pixel quantification of
elemental concentrations. Instead, processing of entire map surfaces using the Bruker Esprit™ software
allows semi-quantitative elemental abundance maps to be created of the sample surface based on a
mapping of the count rate in Regions of Interest of elements (see de Winter and Claeys, 2016; de Winter
et al., 2017b; **Fig. 3**). XRF maps allow for a rapid assessment of the preservation state of original shell
calcite based on variations in Si, Mn, Fe and Sr concentrations and guide the selection of sampling protocols
for further analyses (de Winter and Claeys, 2016; **Fig. 3**). Results of XRF mapping on all 12 *R. diluvianum*
shell cross sections are provided in **S2**.
2.3 Micro-XRF line scans
After XRF mapping, quantitative line scans were measured in growth direction on shell cross sections.
Dwell times of 60 seconds per measurement yielded signal-to-noise ratios sufficient to allow individual
points in line scans to be quantified. This acquisition time was chosen as to provide the optimal compromise
between increasing run time (improved signal/noise ratio; enhanced reproducibility) and increasing the
number of sampling positions (improving sampling density and allowing duplicate measurements) for the
elements Mg, Al, Si, P, S, Ca, Ti, Mn, Fe, Cu, Zn and Sr (TSR and TSA; see discussion in de Winter et al.,
2017b). The sampling density of line scans was 50 μm, adding up to a total of 11056 individual quantitative
XRF spectra measured for this study. Spectra were quantified using the Bruker Esprit software calibrated
using the matrix-matched BAS-CRM393 limestone standard (Bureau of Analyzed samples, Middlesbrough,
UK), after which individual measurements were calibrated offline using 7 matrix-matched certified reference
materials (CCH1, COQ1, CRM393, CRM512, CRM513, ECRM782 and SRM1d), which were treated as
samples (see Vansteenberge et al., in review). $R^2$ values of calibration curves exceeded 0.99 and
reproducibility standard deviations were better than 10% relative to the mean. Even though line scans were
positioned on well-preserved shell calcite based on the XRF map results, a second check was carried out
in which individual points were rejected based on conservative thresholds for diagenetic recrystallization or
detrital contamination ([Ca] < 38 wt%, [Si] > 1 wt%, [Mn] > 200 μg/g or [Fe] > 250 μg/g; [Sr]/[Mn] < 100
mol/mol; see Al-Aasm and Veizer, 1986a; Sørensen et al., 2015). Concentrations of Ca, Mg and Sr in well-
preserved shell sections were used to explore the potential of Mg/Ca and Sr/Ca molar ratios as
paleoenvironmental proxies. Unprocessed results of XRF line scanning are provided in **S3**.
2.4 LA-ICP-MS line scans
Spatially resolved elemental concentrations for Li, B, Mg, Si, P, Ca, Ti, V, Cr, Mn, Fe, Ni, Zn, Rb, Sr, Ba,
Pb and U were calculated from a calibrated transient MS signal recorded during line scanning in the growth
direction (parallel to the XRF line scans) on the shell cross sections using Laser Ablation-Inductively
Coupled Plasma-Mass Spectrometry (LA-ICP-MS). LA-ICP-MS measurements were carried out at the
Atomic and Mass Spectrometry – A&MS research unit of Ghent University (Ghent, Belgium) using a 193
nm ArF*excimer-based Analyte G2 laser ablation system (Teledyne Photon Machines, Bozeman, USA),
equipped with a HelEx 2 double-volume ablation cell, coupled to an Agilent 7900 quadrupole-based ICP-
MS unit (Agilent, Tokyo, Japan). Continuous scanning along shell transects using a laser spot with a
diameter of 25 μm, scan speed of 50 μm/s and detector mass sweep time of 0.5 yielded profiles with a
lateral sampling interval of 25 μm, amounting to a total of 9505 LA-ICP-MS data points gathered. The
aerosol was transported using He carrier gas into the ICP-MS unit via the aerosol rapid introduction system
(ARIS; Teledyne Photon Machines, Bozeman, USA). Elemental concentrations were calibrated using
bracketed analysis runs on US Geological Survey (USGS) BCR-2G, BHVO-2G, BIR-1G, GSD-1G and
GSE-1G and National Institute of Standards and Technology (NIST) SRM612 and SRM610 certified
reference materials. Calcium concentrations (measured via $^{43}$Ca) were used as internal standard for data
normalization and drift correction during the measurement campaign, and Ca concentrations of 38.5 wt%
were assumed for pristine shell carbonate. Coefficients of determination ($R^2$) of a linear model fitted to the
calibration curves were better than 0.99 and the standard deviation of reproducibility for elemental
concentrations was better than 5% relative to the mean value. Individual LA-ICP-MS measurements were
inspected for diagenetic alteration or contamination by detrital material using the same thresholds as used
for XRF data (see above). LA-ICP-MS and μXRF measurements were combined to cover a wider range of





elements, since some elements (e.g. S and Sr) were measured more reliably using µXRF, while others
(e.g. Li or Ba) could only be determined using LA-ICP-MS. Concentrations of Li, Mg, and Sr were used to
explore the potential of Mg/Li and Sr/Li molar ratios as proxies for paleoenvironmental change.
Unprocessed results of LA-ICP-MS line scans are provided in **S4**.
2.5 Isotope Ratio Mass Spectrometry
A transect of powdered samples (±200 µg) was sampled for Isotope Ratio Mass Spectrometry (IRMS)
analysis in growth direction along well-preserved foliated calcite (**Fig. 3**) in the five largest of the twelve *R.*
*diluvianum* shells (*R. diluvianum* 1-5; see **Fig. 2**) using a microdrill (Merchantek/Electro Scientific Industries
Inc., Portland (OR), USA) equipped with a 300 µm diameter tungsten carbide drill bit, coupled to a
microscope (Leica GZ6, Leica Microsystems GmbH, Wetzlar, Germany). A total of 531 IRMS samples were
taken at an interspacing of 250 µm. Stable carbon and oxygen isotope ratios ($\delta^{13}C$ and $\delta^{18}O$) were
measured in a NuPerspective IRMS equipped with a NuCarb carbonate preparation device (Nu
Instruments, UK). The sample size (50-100 µg) allowed duplicate measurements to be carried out regularly
to assess reproducibility. Samples were digested in 104% phosphoric acid at a constant temperature of
70°C and the resulting $CO_2$ gas was cryogenically purified before being led into the IRMS through a dual
inlet system. Isotope ratios were corrected for instrumental drift and fractionation due to variations in sample
size and the resulting values are reported in per mille ratios calibrated to the Vienna Pee Dee Belemnite
standard (‰VPDB) using repeated measurements of the IA-603 stable isotope standard (International
Atomic Energy Agency, Vienna, Austria). Reproducibility of $\delta^{18}O$ and $\delta^{13}C$ measurements on this standard
were better than 0.1‰ and 0.05‰ (1σ; N=125) respectively. All stable isotope analysis results are provided
in **S5** and plots of stable isotope and trace element records from all shells are shown in **S6**.
2.6 Growth and age modelling
Stable oxygen isotope curves measured in *R. diluvianum* were used to produce age models for the growth
of the shell using a bivalve growth model written in MatLab (Mathworks, Natick, MA, USA) which simulates
$\delta^{18}O$ curves using a combination of a growth sinusoid and a temperature sinusoid to fit the $\delta^{18}O$ data (Judd
et al., 2018). This simulation model was modified to calculate its temperatures based on calcite $\delta^{18}O$
(following Kim and O'Neil, 1997) rather than from the aragonite $\delta^{18}O$-temperature relationship used in the
original approach (after Grossman and Ku, 1986; see Judd et al., 2018). A value of -1.0‰ VSMOW was
assumed for $\delta^{18}O$ of Campanian ocean water (Thibault et al., 2016). Additional minor modifications in the
source code allowed results of intermediate calculation steps in the model to be exported. The modified
Matlab source code is provided in **S7**. Note that this model assumes that the shape and absolute value of
$\delta^{18}O$ curves depend solely on water temperature and growth rate (ignoring changes in sea water $\delta^{18}O$), and
that a modelled year contains 365 days by construction (while this number should be slightly larger in the
Late Cretaceous; e.g. Meyers and Malinverno, 2018; de Winter et al., in review). Nevertheless, shell
chronologies reconstructed from seasonal patterns in $\delta^{18}O$ should still be reliable regardless of their origin.
Uncertainties on modelled temperature curves were derived by propagating the measurement uncertainty
on $\delta^{18}O$. Age models thus obtained for shells *R. diluvianum* 1-5 were used to align all proxy records on a
common time axis. Age models for *R.diluvianum* 6-12 were constructed by extrapolating relationships
between modelled seasonality and microstructures and trace element concentrations observed in *R.*
*diluvianum* 1-5. Simultaneously deposited microstructural features in shell cross sections (see **Fig. 3**) were
used to determine the actual dorsoventral height of the shells at different ages, linking shell height to the
age and allowing the construction of growth curves for all twelve *R. diluvianum* shells.
2.7 Strontium isotopic analysis
Samples (26 mg) for strontium isotopic analysis were obtained by drilling the well-preserved foliated calcite
in all shells using a Dremel 3000 dental drill with a 0.5 mm tungsten carbide drill bit. Calcite samples were
dissolved in subboiled concentrated (14 M) nitric acid ($HNO_3$) at 120°C and left to dry out at 90°C overnight,
after which the residue was redissolved in 1 M $HNO_3$. Strontium in the samples was purified following the
ion-exchange resin chromatography method detailed in Snoeck et al. (2015). The $^{87}Sr/^{86}Sr$ of purified Sr
samples were determined using a Nu Plasma (Nu Instruments Ltd, Wrexham, UK) multi-collector (MC) ICP-
MS unit in operation at the Université Libre de Bruxelles (ULB). During the measurement run, repeated
analyses of NIST SRM987 standard solution yielded a ratio of 0.710250 ± 40 (2 SD; N = 14), statistically
consistent with the literature value of 0.710248 ± 5.8 (2 s.e.; McArthur et al., 2001; Weis et al., 2006). All



results were corrected for instrumental mass discrimination by internal normalization and normalized to the
literature value of NIST SRM987 (0.710248) through a standard-sample bracketing method. For each
sample, $^{87}Sr/^{86}Sr$ are reported with a 2 standard deviations uncertainty (**S8**).
***Figure 4:*** Plot showing the results of Sr-isotopic analyses with error bars (2 SD) plotted on the Sr-isotope curve of
McArthur et al. (2016; top of image). Numbers below the error bars indicate sample number. Measurements from the
12 specimens of *R. diluvianum* are represented by parallelograms in different shades of blue which correspond to the
graph below. The probability distribution curves in the lower pane show the distribution of uncertainty on each Sr-



isotope measurement as well as the uncertainty on the Sr-isotope curve propagated to the age domain (colors of
individual shells are the same as in **Fig. 2**). Insert shows schematically how uncertainties of the isotope measurements
as well as the isotope curve are propagated into the age domain. The black curve shows the total uncertainty distribution
function compiled from the 12 individual measurements following Barlow (2004), with the combined age estimate
including uncertainty (2 SD) shown above.
2.8 Strontium isotope dating
*R.* diluvianum specimens were independently dated by comparing $^{87}Sr/^{86}Sr$ values measured in the
samples with the Sr-isotope curve in the 2016 Geological Timescale (McArthur et al., 2016). Uncertainties
in $^{87}Sr/^{86}Sr$ measurements were propagated into dates by finding the closest date of the mean $^{87}Sr/^{86}Sr$
value as well as the dates of the minimum (-2σ) and maximum (+2σ) $^{87}Sr/^{86}Sr$ values by linearly interpolating
ages in the $^{87}Sr/^{86}Sr$ curve matching the measured $^{87}Sr/^{86}Sr$ value, including the uncertainty estimated on
the Sr-isotope curve itself. A composite age for the Ivö Klack deposits was obtained by combining the age
uncertainty distributions of the individually dated $^{87}Sr/^{86}Sr$ samples into a single age. Due to the non-linear
shape of the $^{87}Sr/^{86}Sr$ curve, uncertainties on the $^{87}Sr/^{86}Sr$ ages were asymmetrical. Since no mathematical
solution exists for the combination of asymmetric uncertainties, the asymmetric uncertainty on the total age
has to be approximated through maximum likelihood estimation using the combined log likelihood function
(Barlow, 2003). The approximation of the total uncertainty of combined $^{87}Sr/^{86}Sr$ dating results in this study
was carried out using the mathematical approach of Barlow (2004) in R (R Core Team, 2013; Roger Barlow,
personal communication; code available on https://zenodo.org/record/1494909). The uncertainty interval of
the composite age is represented by 2 times the standard error (~95.5% confidence level). A plot of the
uncertainty distributions of the individual specimens and the total uncertainty distribution is shown in **Fig.**
**4**. Raw $^{87}Sr/^{86}Sr$ data is provided in **S8**.

**3. Results**
3.1 Dating
The compilation of $^{87}Sr/^{86}Sr$ results from 12 specimens of *R. diluvianum* (**Fig. 4**) shows how age estimates
from individual specimens have considerable uncertainties (standard deviations around 1 Myr, see **S8**), yet
the uncertainty on the composite age is significantly smaller. The composite age for the Ivö Klack deposits
is 78.14 Ma (±0.26; 2 standard errors). This result places the age of the Ivö Klack deposits close to the
early/late Campanian boundary when applying a twofold division of the Campanian and in the middle
Campanian when applying a threefold division scheme (Ogg et al., 2016). This age estimate is similar to
the age obtained when plotting the *B. mammilatus* zone on the recent integration schemes of the
Campanian (Wendler, 2013). Earlier estimates (Christensen, 1997; Surlyk and Sørensen, 2010; Sørensen
et al., 2015) yielded ages about 2-4 Myr older (80-82 Ma), but those relied on presently outdated and partly
incorrect age models.
3.2 Shell structure and preservation
A combination of high-resolution color scans, SEM images and µXRF mapping of shell cross sections
reveals that *R. diluvianum* shells consist of thin layers of dark, foliated calcite, interwoven with lighter, more
porous carbonate layers. The latter are characterized by higher concentrations of Mn, Fe and Si and lower
Sr concentrations (**Fig. 3**). Foliated calcite layers are more densely packed on the inside of the shell,
especially in the region of the adductor muscle scar (**Fig. 3**). They are characterized by high Sr
concentrations and low concentrations of Mn, Fe and Si (**Fig. 3**; **S2**). Foliated layers are also densely
packed at the shell hinge. Further away from the shell hinge and the inside of the valve, porous carbonate
layers become more dominant. In these regions, µXRF mapping also clearly shows that detrital material
(high in Si and Fe) is often found between the shell layers. SEM images show that the shell structure of *R.*
*diluvianum* strongly resembles to that of modern oyster species, as described in previous studies (Carriker
et al., 1979; Surge et al., 2001; Ullmann et al., 2010; 2013; Zimmt et al., 2018). The major part of the shell
consists of (foliated and porous) calcite structures, which were sampled for chemical analyses in this study.
As in modern oyster species, aragonite may originally have been deposited on the resilium of the shell, but
this region is not considered for analyses (Stenzel, 1963; Carriker et al., 1979; Sørensen et al., 2012). Close





similarities with modern oysters allow to infer that shell growth in *R. diluvianum* occurred in a similar way
as it does in modern oyster species like *Ostrea edulis*, *Crassostrea virginica* and *C. gigas*. This extrapolation
allows to estimate the total shell height from microstructural growth markers (**Fig. 3**; following Zimmt et al.,
2018), linking growth to changes in shell chemistry. It also allows chemical changes in the shell to be
interpreted in terms of environmental changes by applying calibration curves for trace element proxies that
were previously established for modern oyster species (e.g. Surge and Lohmann, 2008; Ullmann et al.,
2013; Mouchi et al., 2013; Dellinger et al., 2018).
3.3 Trace element analyses results
The combination of μXRF, LA-ICP-MS and IRMS analyses on *R. diluvianum* shells resulted in multi-proxy
records of changes in Mg/Ca, Sr/Ca (μXRF), Mg/Li, Sr/Li (LA-ICP-MS), $\delta^{13}$C and $\delta^{18}$O (IRMS, only for shells
1-5, see **Fig. 2**). All chemical analyses were carried out on the dense foliated calcite exposed in cross
sections close to the inner edge of the shell valve (**Fig. 3**). High-resolution color scans and detailed
recording of sampling positions allowed these records to be plotted on a common axis (see **S6**). In **Fig. 5**,
results of chemical analyses of *R. diluvianum* specimens (including diagenetic parts) are compared with
data from three other mollusk taxa (*Belemnellocamax mammillatus*, *Acutostrea incurva* and radiolithid
rudists) from Ivö Klack (Sørensen et al., 2015), as well as data from extant oysters (Rucker and Valentine,
1961; Surge et al., 2001; Ullmann et al., 2013). **Figure 5** shows that stable isotope ratios of the rudist and
oyster shells overlap, while belemnites are characterized by much lower $\delta^{13}$C and heavier $\delta^{18}$O values. This
suggests that $\delta^{13}$C in belemnite rostra are affected by vital effects while heavier $\delta^{18}$O values of the
belemnites suggest that these animals lived most of their life span in a different environment than the
bivalves (deeper waters), as previously suggested by Sørensen et al. (2015). By contrast, stable isotope
ratios recorded in the bivalve shells overlap and match the isotope ratios measured in Campanian chalk
deposited in the neighboring Danish Basin (Thibault et al., 2016). Multi-proxy analysis revealed periodic
variations in stable isotope and trace element ratios (see **Fig. 6**). The amplitudes of these variations plotted
in **Fig. 5** show that Mg and Sr concentrations measured in all three fossil bivalve taxa are similar, while
concentrations in the belemnite rostra are much higher. Finally, plots of Sr and $\delta^{18}$O against Mn
concentrations demonstrate that diagenetic alteration (evident from elevated Mn concentrations) reduces
the Sr concentration in carbonate of all four taxa. Stable oxygen isotope ratios of the shells are affected to
a lesser degree. The vast majority of measurements in all four taxa show very little signs of diagenetic
alteration, with most measurements characterized by low (< 100 μg/g) Mn concentrations (**Fig. 5**).



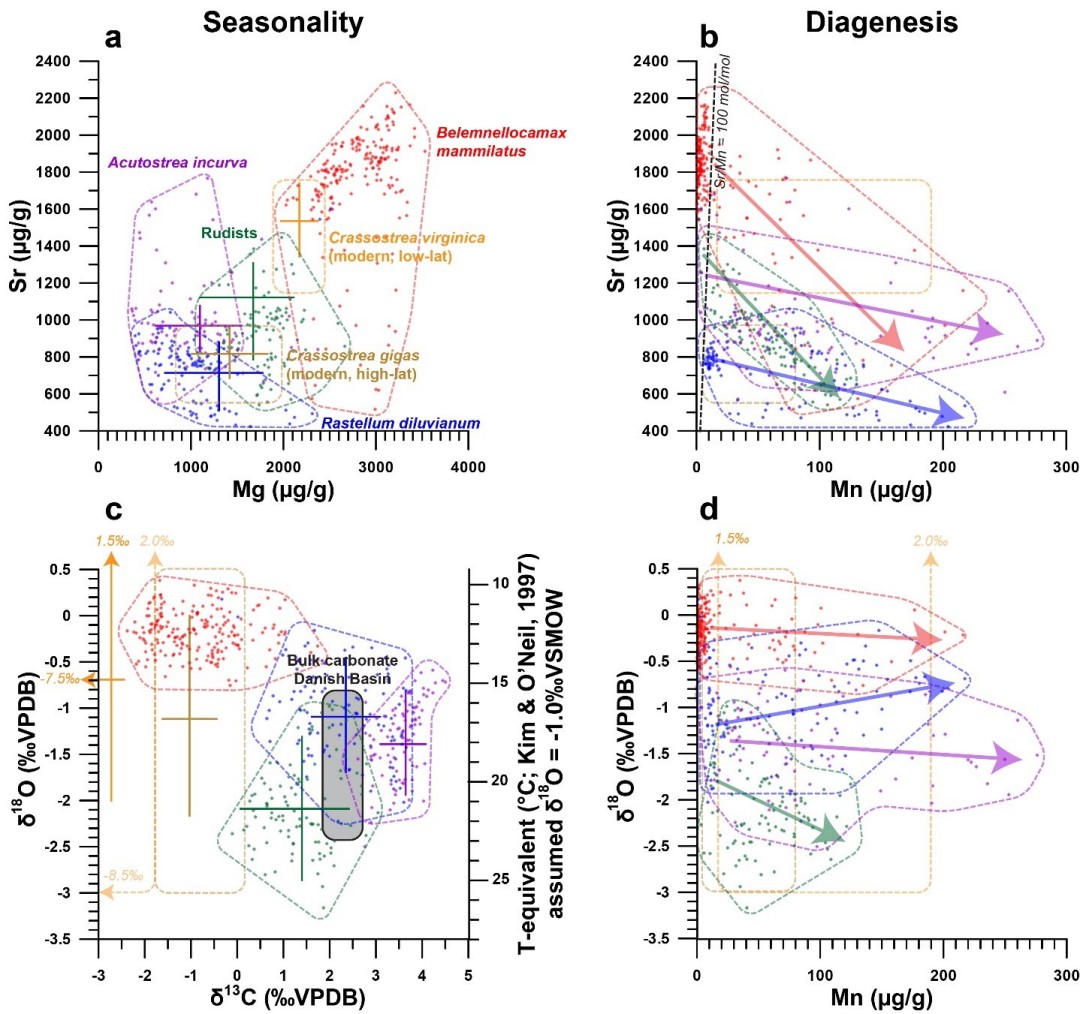

*Figure 5:* Cross plots summarizing the results of trace element and stable isotope analyses of the oysters *R. diluvianum* (blue), *A. incurve* (purple), associated rudist bivalves (green) and the belemnite *B. mammilatus* (red, after Sørensen et al., 2015) from the Kristianstad basin. Results in modern *C. gigas* (grey/black; Ullmann et al., 2013) and *C. virginica* (orange/yellow; Rucker and Valentine, 1961; Surge et al., 2001) oysters are plotted for comparison. Points indicate individual data points, drawn polygons illustrate the range of the data and crosses indicate the extent of seasonality (if present). (**a**) Strontium concentrations plotted against magnesium concentrations. (**b**) Strontium concentrations plotted against manganese concentrations. Arrows indicate the interpreted direction of diagenetic alteration and the black dashed line shows the Sr/Mn diagenesis threshold proposed by Sørensen et al. (2015; 100 mol/mol). (**c**) $\delta^{18}$O plotted against $\delta^{13}$C. Grey area indicates the range of stable isotope ratios measured in Campanian chalk deposits from the nearby Danish Basin by Thibault et al. (2016) (**d**) $\delta^{18}$O plotted against manganese concentrations, with arrows indicating proposed direction of diagenetic alteration.

3.4 Stable isotope records

Records of $\delta^{13}$C and $\delta^{18}$O in the growth direction through *R. diluvianum* shells exhibit periodic variations. These variations are much more regular in $\delta^{18}$O records, which show extreme values of -3‰ up to 0‰ VPDB. Some shells, such as *R. diluvianum* 3 (**Fig. 6**), exhibit longer term trends on which these periodic variations are superimposed. These trends suggest the presence of multi-annual cyclicity with a period in the order of 10-20 years, but the length of *R. diluvianum* records (max. 10 years) is smaller than the estimated period of these changes and is therefore not sufficient to statistically validate the presence of this



cyclicity. The extreme values in $\delta^{18}O$ records translate to temperatures in the range of extremes of 12°C to
26°C when assuming a constant $\delta^{18}O_{seawater}$ value of -1.0‰ (e.g. Thibault et al., 2016) and applying the
temperature relationship of Kim and O'Neil (1997). Carbon isotope ratios ($\delta^{13}C$) do not always follow the
same trends as $\delta^{18}O$ records. In many parts of *R. diluvianum* shells, there is a clear covariation between
the two isotope ratios, suggesting $\delta^{13}C$ is affected by seasonal changes. However, in other parts this
correlation is less clear, suggesting that other (non-seasonal) factors play a role in determining the $\delta^{13}C$ of
shell material. Superimposed on these changes, a statistically significant ontogenetic trend can be
discerned in the $\delta^{13}C$ records of 10 out of 12 shells. However, the scale and direction of these trends do
not seem consistent between shells.
3.5 Age models
Modelling the growth of *R. diluvianum* bivalves from seasonal variations in $\delta^{18}O$ profiles yielded age models,
growth rate estimates and reconstructions of water temperature variations during the lifetime of the bivalves.
Due to the clear seasonal patterns in $\delta^{18}O$ records (**Fig. 6**), modelled $\delta^{18}O$ profiles closely approximated
the measured $\delta^{18}O$ profiles (total $R^2$ = 0.86, N = 412, see **S9**), lending high confidence to shell age models.
Modelling allowed all proxies measured in the shells of *R. diluvianum* to be plotted against shell age, clearly
revealing the influence of seasonal variations in environmental parameters on shell chemistry (**S10**). When
plotting all proxies on the same time axis, clear ontogenetic trends emerge in Mg/Li, Sr/Li and $\delta^{13}C$ in nearly
all specimens of *R. diluvianum*. Trends and variations in Mg/Li and Sr/Li are strongly correlated, suggesting
that variation in both these trace element ratios is largely driven by variations in Li concentrations. Linear
regression was applied to isolate ontogenetic trends in $\delta^{13}C$ and Li/Ca ratios (**S11-S12**). While most of these
ontogenetic trends are statistically significant (p < 0.05), they are highly variable between specimens, both
in terms of direction and magnitude. The distribution of slopes of ontogenetic trends in Li/Ca and $\delta^{13}C$
cannot be distinguished from random variation (see **Table 1**). Therefore, no predictable ontogenetic trends
were found for $\delta^{13}C$ and Li-proxies in *R. diluvianum* shells.

| | Li/Ca | | | $\delta^{13}C$ | | |
|---|---|---|---|---|---|---|
| | slope (mol/(mol*yr)) | R2 | p-value | slope (‰/yr) | R2 | p-value |
| *R. diluvianum 1* | -1.29E-06 | 0.053 | 4.32E-08 | 0.346 | 0.426 | 8.86E-07 |
| *R. diluvianum 2* | 3.74E-07 | 0.101 | 2.68E-05 | 0.169 | 0.440 | 8.19E-08 |
| *R. diluvianum 3* | 3.86E-07 | 0.004 | 5.32E-03 | -0.004 | 0.001 | 8.09E-01 |
| *R. diluvianum 4* | -1.07E-06 | 0.025 | 8.78E-04 | 0.023 | 0.009 | 3.99E-01 |
| *R. diluvianum 5* | -1.94E-06 | 0.030 | 6.30E-14 | 0.136 | 0.492 | 5.53E-11 |
| *R. diluvianum 6* | -2.32E-06 | 0.117 | 8.75E-15 | | | |
| *R. diluvianum 7* | -7.49E-07 | 0.029 | 4.77E-02 | | | |
| *R. diluvianum 8* | -1.19E-07 | 0.003 | 2.90E-01 | | | |
| *R. diluvianum 9* | -4.63E-07 | 0.010 | 5.65E-02 | | | |
| *R. diluvianum 10* | 1.59E-06 | 0.015 | 1.61E-02 | | | |
| *R. diluvianum 11* | -1.87E-06 | 0.199 | 4.25E-12 | | | |
| *R. diluvianum 12* | -4.55E-07 | 0.003 | 4.19E-01 | | | |

| | | | | |
|---|---|---|---|---|
| p($\chi^2$) | 0.976 | p($\chi^2$) | | 1.000 |
| p($\chi^2$) weighed by R2 | 0.976 | p($\chi^2$) weighed by R2 | | 1.000 |
| p($\chi^2$) weighed by p-value | 0.961 | p($\chi^2$) weighed by p-value | | 0.998 |


**Table 1:** *Overview of the slopes of ontogenetic trends in Li/Ca and $\delta^{13}C$ records. P-values on the bottom of the table show that the*
*distribution of slopes is statistically indistinguishable from random.*



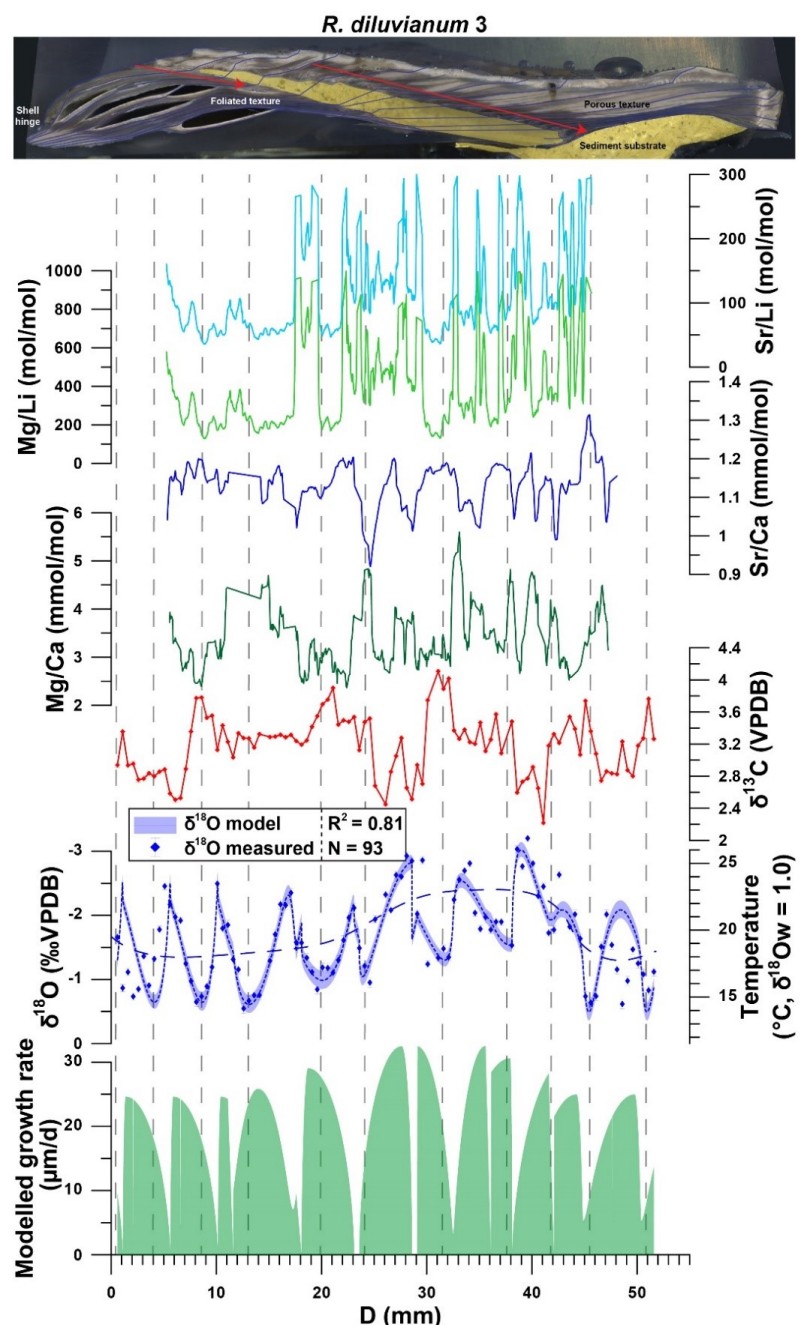

**Figure 6:** Example of multi-proxy records measured in *R. diluvianum* specimen 3 plotted against distance in growth direction (see image on top and **Fig. 3** for reference). From top to bottom, records of Sr/Li (light blue), Mg/Li (light green), Sr/Ca (dark blue), Mg/Ca (dark green), $\delta^{13}C$ (red), $\delta^{18}O$ (blue dots with error bars) and modelled growth rate (light green fill) are plotted. The shaded blue curve plotted underneath the $\delta^{18}O$ record illustrates the result of growth and $\delta^{18}O$ modelling and its propagated error (vertical thickness of curve, 2SD). The dashed blue curve plotted on top of the $\delta^{18}O$ record shows the observed multi-annual trend in the data.






3.6 Trace element seasonality
A comparison of the amplitude of periodic variations in Mg/Ca, Sr/Ca, Mg/Li and Sr/Li in 12 *R. diluvianum*
shells (**Fig. 7**), together with a tentative interpretation in terms of temperature seasonality, reveals that it is
not straightforward to apply the transfer functions previously proposed for these proxies on fossil bivalve
shells. Results reveal a strong positive inter-shell correlation between Sr/Li and Mg/Ca ($R^2$ = 0.76) and
between Sr/Li and Mg/Li ($R^2$ = 0.93), while positive correlations between Sr/Ca and Mg/Ca ($R^2$ = 0.19) as
well as between Sr/Ca and Mg/Li ($R^2$ = 0.20) are weak. The Mg/Li temperature regressions based on
benthic foraminifera (Bryan and Marchitto, 2008) yield unrealistically high-water temperatures (> 50°C),
presumably due to typically lower Mg concentrations in foraminifera compared to bivalves (Yoshimura et
al., 2011). The Mg/Ca and Sr/Li temperature relationships (Surge and Lohmann, 2008; *C.* virginica; and
Füllenbach et al., 2015; *Cerastoderma edule*; respectively) and a Mg/Li temperature regression based on
the calcitic bivalve *Mytilus edulis* (Dellinger et al., 2018) yield temperatures in the same range as those
reconstructed from local bulk carbonate stable isotope measurements (10-20°C; e.g. Thibault et al., 2016),
but Sr/Li-based temperatures display a pattern opposite to those based on Mg-proxies. This seems to
suggest that, if trace element concentrations in *R. diluvianum* are linked to temperature, the temperature
relationship of Mg-based proxies and the Sr/Li proxy are discordant and cannot both be applicable to *R.*
*diluvianum*. These results raise difficulties similar to those that arose in earlier attempts to apply trace
element ratios for water temperature reconstructions in fossil mollusks (Steuber, 1999; Weiner and Dove,
2003; de Winter et al., 2017a). The interpretation of these records is further complicated by large intra-
specific variability in the incorporation of Mg into biogenic carbonates (e.g. Schöne et al., 2010) and the
lack of constraints of seawater compositions in the Late Cretaceous (e.g. Stanley and Hardie, 1998;
especially with respect to Li concentrations). It shows that trace element ratios in these shells can only be
interpreted with some degree of confidence when combined with stable isotope records from shells of the
same setting and species.

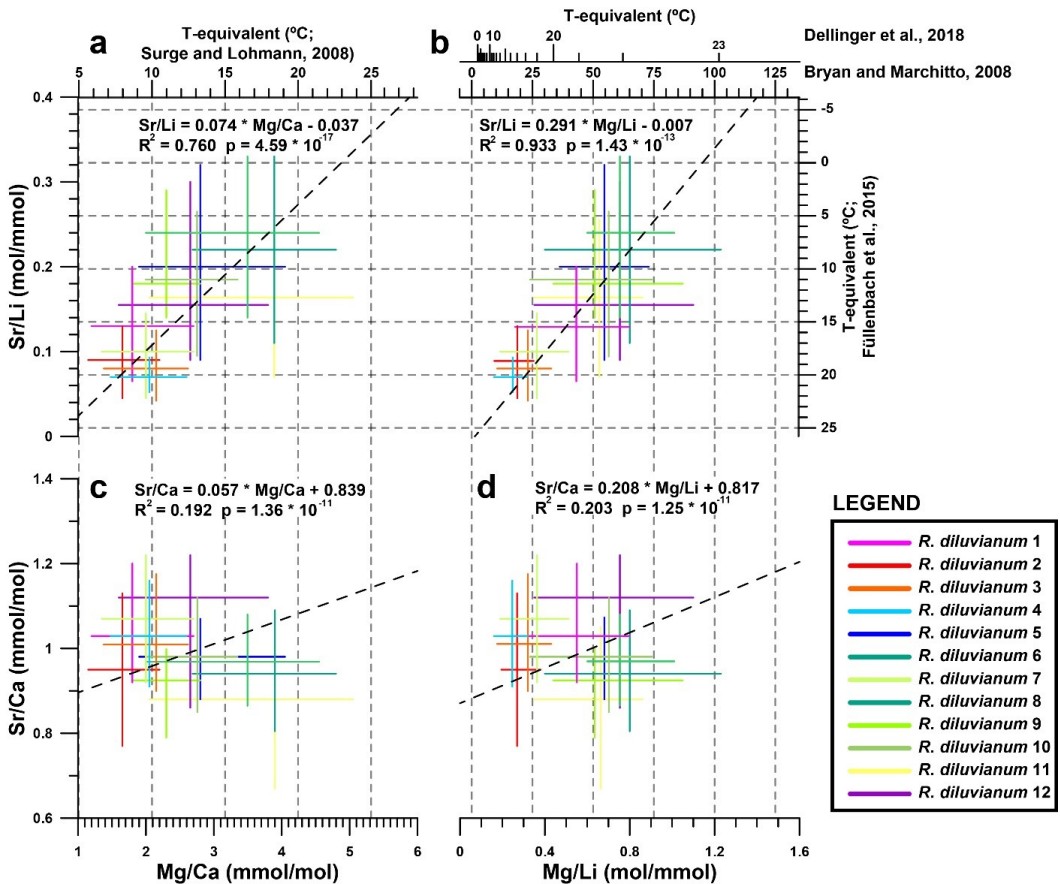

**Figure 7:** Cross plots showing the extent of interpreted seasonality observed in records of four trace element proxies in all 12 *R. diluvianum* specimens. Colors of lines of individual shells correspond to colors indicated in **Fig. 2**. Temperature conversions from previously published regressions of the proxies with temperature are shown on opposite axes with grey dashed lines corresponding to major tick marks on the temperature scale (**a**) Sr/Li plotted against Mg/Ca showing a strong significant intra-shell correlation. (**b**) Sr/Li plotted against Mg/Li showing a strong significant intra-shell correlation due to dominant variations in Li concentration. Note that two different Mg/Li temperature calibrations were explored. (**c**) Sr/Ca plotted against Mg/Ca showing weak but significant intra-shell correlation. (**d**) Sr/Ca plotted against Mg/Li showing a weakly significant intra-shell correlation. Data for this plot is found in **S13**.

### 3.7 Temperature seasonality

The seasonal variation in all specimens of *R. diluvianum* was aligned and stacked relative to shell age models (**Fig. 8**). This composite stack shows that the seasonal temperature range in Ivö Klack during the late early Campanian was between 16°C and 21°C when assuming constant seawater $\delta^{18}O$. Modelled growth rates in *R. diluvianum* peak near the end of the low temperature season and average growth rates are lowest shortly after the temperature maximum (**Fig. 8**). This phase shift between temperature and growth rate could indicate that growth in *R. diluvianum* in this setting was not limited by low temperatures, as observed in modern mid- to high-latitude oysters (Lartaud et al., 2010). High temperature extremes (>25°C) may have slowed or stopped growth, as recorded in modern low latitude settings (Surge et al., 2001), but $\delta^{18}O$-seasonality suggests that these temperatures were not common at the Ivö Klack locality. Mg/Ca ratios in *R. diluvianum* exhibit a clear seasonal pattern, which is inversely correlated with temperature, while Mg/Ca ratios in foliate calcite of modern oysters show opposite correlation with temperature (Surge and Lohmann, 2008; Mouchi et al., 2013) or exhibit no correlation at all (Ullmann et al., 2013). Sr/Ca ratios in *R. diluvianum* are positively correlated with seasonal temperature variations. Mg/Li and Sr/Li ratios show no correlation with temperature. Instead, both proxies display elevated values both





directly before and after seasonal temperature maxima (**Fig. 8**). Finally, $\delta^{13}C$ values exhibit no observable
relationship with temperature seasonality.

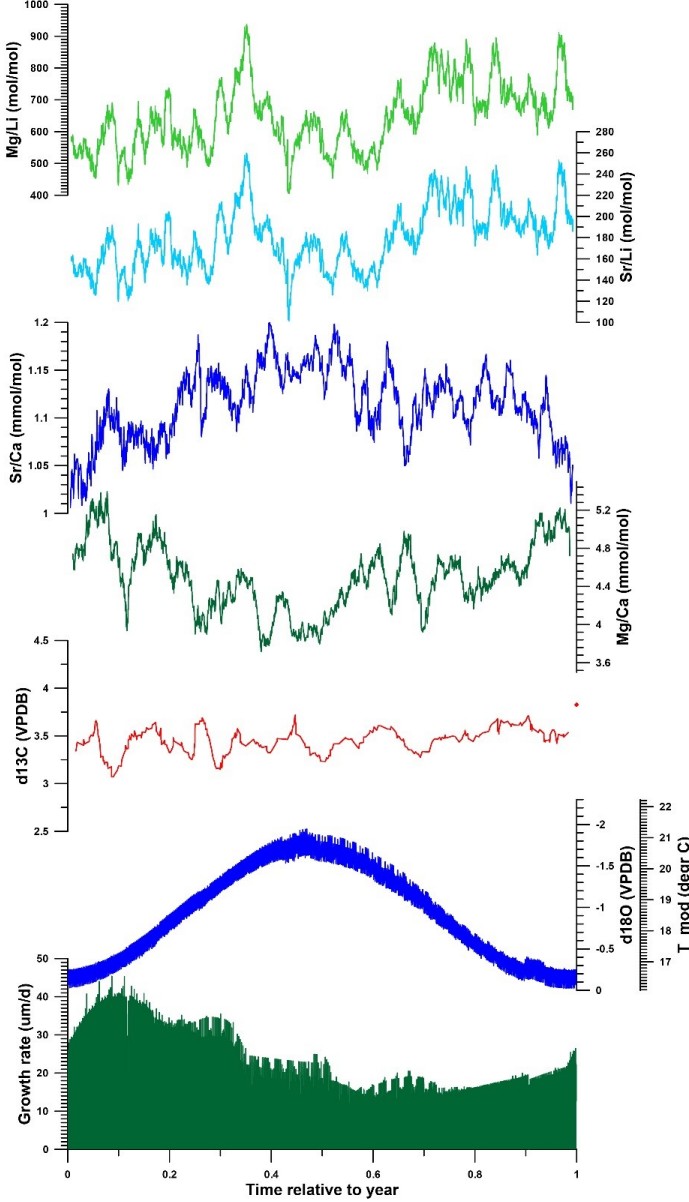

***Figure 8***: Composite of multi-proxy records from all *R. diluvianum* shells stacked and plotted on a common time axis of 1 year to illustrate the general phase relationships between various proxies in the shells. Records were colored as in **Fig. 6**. Annual stacks plotted in this figure were produced/obtained by applying age models on all multi-proxy records, plotting all results against their position relative to the annual cycle and applying 20 point moving averages.


3.8 Shell growth



Plots of modelled shell height against age allow to compare growth patterns of individual *R. diluvianum*
(**Fig. 9**). Individual growth curves clearly converge to a general growth development curve for the species.
Considering that the isotope transects used to establish these growth curves were measured in different
stages of life in different specimens (large age variation), individual growth curves are remarkably similar.
The growth of *R. diluvianum* takes the typical shape of the asymptotic Von Bertalanffy curve, in which shell
height ($H_t$) development with age ($t$) is related to a theoretical adult size $H_{max}$ and a constant $k$ in the
equation: $H_t[mm] = H_{max} * \left(1 - e^{k*(t[yr]-t_0)}\right)$, with $t_0$ representing the time at which the growth period
started (always zero in this case; Von Bertalanffy, 1957). When this formula is regressed over all modelled
growth data of all shells (1 data point per day, 15146 points in total), the fit with an $H_{max}$ of ±120.3 mm and
a K value of ±0.32 is very good ($R^2 = 0.79$; see **Fig. 9**).

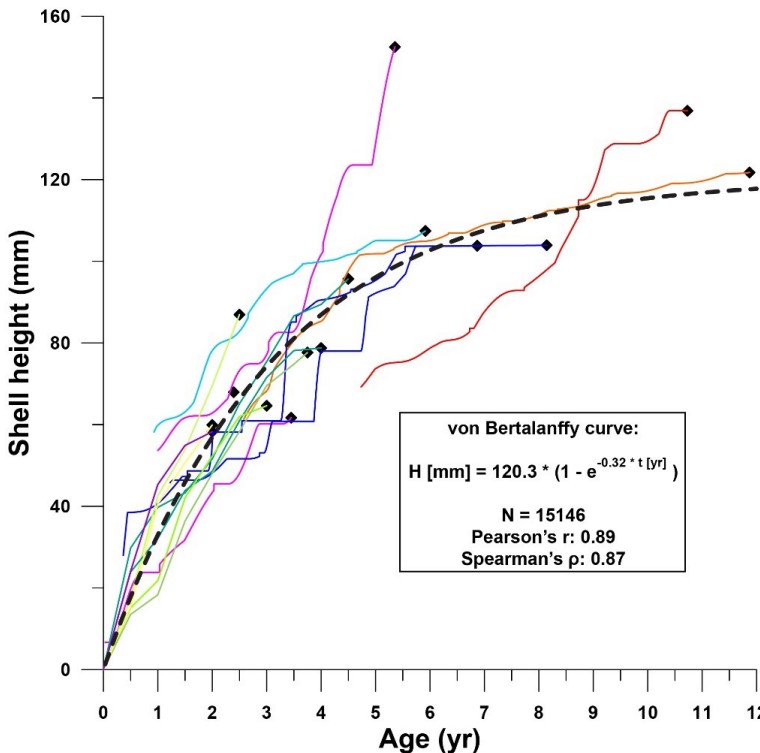


*Figure 9*: Shell height plotted against age for all *R. diluvianum* records (see **Fig. 7** for color legend of lines representing individuals).
The similarity between growth curves of different specimens allows a Von Bertalanffy curve to be fitted to the data with high confidence.
Sinusoidal patterns superimposed on all growth curves are caused by seasonal variability in growth rate (see **Fig. 6** for an example).
Data found in **S9**.
3.9 Statistics in seasonal growth and ecology
The seasonality stack of growth rates shown in **Fig. 8** suggests a potential year-round growth in *R.*
*diluvianum*, but this is a bias induced by the way the annual stack is plotted. To better understand the
growth and life history of *R. diluvianum* oysters, it is important to consider the variability between individual
years of growth in the different individuals. Using oxygen isotope records, year-long "seasonal" cycles and
subsequently derived growth rates from our 12 specimens of *R. diluvianum*, we isolated statistics of
individual growth seasons in order to visualize the potential relationship between growth rate, temperature
and time of year (**Fig. 10**). The onset and end of each year correspond to maxima in $\delta^{18}O$ values (minima
in temperatures). Isolating all 58 individual growth years in specimens used in this study based on the
temperature seasonality modelled on $\delta^{18}O$ records allowed a comparison of statistics such as seasonal
minima and maxima in growth, the length of the growth season and the extent of seasonality to be evaluated



(**Fig. 10**). The onset of the first growth year in each shell at its precise position relative to the seasonal
temperature cycle showed in which season spawning occurred (**Fig. 10c**). Finally, evaluation of the
distribution of growth maxima and minima along the seasonal cycle and regression analyses between these
parameters reconstructed from the growth models shed light on the relationships between growth
parameters in *R. diluvianum* and seasonality All data used to create plots in **Fig. 10** is provided in **S14**.

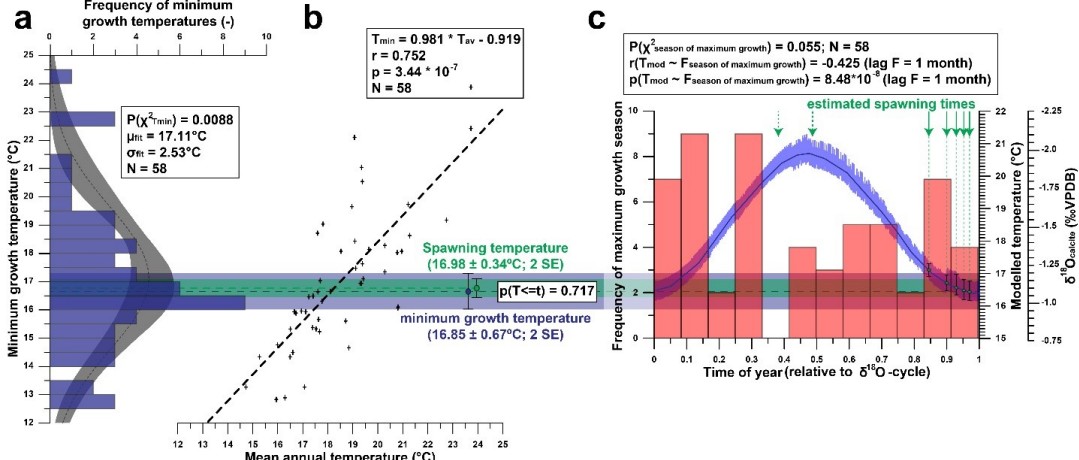


***Figure 10:*** Overview of statistical evaluation of growth parameters of *R. diluvianum* derived from age modelling in shells 1-5. (**a**)
Histogram of minimum temperatures of growth in *R. diluvianum* showing that the temperature on which growth slows coincides with
that of the spawning season (p = 0.717). (**b**) Strong significant positive correlation between MAT and temperature of the slowest
growth season shows that the season of minimum growth is not strictly forced by minimum temperatures but rather by timing relative
to the annual $\delta^{18}$O cycle. (**c**) Histogram of the season of maximum growth relative to the $\delta^{18}$O seasonality cycle shows no significant
concentration towards a favorable growing season while moments of first growth (spawning) are significantly concentrated towards
the low-$\delta^{18}$O season.

| N = 58 | Total annual growth (μm) | | Maximum growth rate (μm/d) | | Length of season (d) | | Minimum growth temperature (°C) | | Temperature seasonality (°C) | | Average temperature (°C) | |
|---|---|---|---|---|---|---|---|---|---|---|---|---|
| Temperature seasonality (°C) | $R^2$ | 0.024 | $R^2$ | 0.053 | $R^2$ | 0.403 | | | | | | |
| | p | $2.16*10^{-11}$ | p | $6.73*10^{-10}$ | p | $2.15*10^{-22}$ | | | | | | |
| Average temperature (°C) | $R^2$ | 0.020 | $R^2$ | 0.027 | $R^2$ | 0.008 | $R^2$ | 0.565 | | | | |
| | p | $2.29*10^{-11}$ | p | $6.95*10^{-7}$ | p | $2.87*10^{-21}$ | p | $3.44*10^{-7}$ | | | | |
| Age (yr) | $R^2$ | 0.000 | $R^2$ | 0.062 | $R^2$ | 0.002 | $R^2$ | 0.002 | $R^2$ | 0.059 | $R^2$ | 0.000 |
| | p | $1.11*10^{9}$ | p | $9.74*10^{-12}$ | p | $1.59*10^{-22}$ | p | $1.05*10^{-30}$ | p | $4.59*10^{-1}$ | p | $1.09*10^{-35}$ |


**Table 2**: Overview of statistical evaluation of growth parameters of *R. diluvianum* derived from age modelling in shells 1-5. Coefficients
of determination ($R^2$) and p-values were determined for relationships between temperature seasonality, average temperature, age of
the bivalve, length of the season, minimum growth temperatures and annual average and maximum growth rates. Values in green
indicate strong correlations while values in red indicate the absence of a correlation. Data reported in **S14**.

## 4. Discussion

4.1 Preservation
The relative lack of burial and tectonic activity in the Kristianstad Basin has provided ideal circumstances
for the nearly immaculate preservation of *R. diluvianum* shells in the Ivö Klack locality (Kominz et al., 2008;
Surlyk and Sørensen, 2010). The excellent state of these shells is evident by the preservation of original
(porous and foliated) microstructures that closely resemble those reported for several species of modern
ostreid shells (Carriker et al., 1979; Surge et al., 2001; Ullmann et al., 2010; 2013; Zimmt et al., 2018; **Fig.**
**2-3**). High magnification SEM images demonstrate the excellent preservation of foliated and vesicular
calcite structures in *R. diluvianum* shells (**Fig. 3b-d**). The preservation state of *R. diluvianum* shells meets
the criteria for robust stable isotope analysis set by Cochran et al. (2010). MicroXRF mapping reveals that



the foliated calcite in the shells is characterized by high Sr concentrations and low concentrations of Mn,
Fe and Si, elements which are generally associated with diagenetic alteration (e.g. Brand and Veizer, 1980;
Al-Aasm and Veizer, 1986a; Immenhauser et al., 2005; **Fig. 3b-h**). Typically, a Mn concentration threshold
of 100 μg/g is applied below which Cretaceous low-magnesium carbonates are assumed suitable for
chemical analysis (Steuber et al., 2005; Huck et al., 2011). Strontium concentrations above 1000 μg/g have
also been used as markers for good preservation, since diagenetic processes can cause strontium to leach
out of carbonates (e.g. Brand and Veizer, 1980; Huck et al., 2011; Ullmann and Korte, 2015). Therefore,
previous studies of belemnites in Kristianstad Basin proposed a molar Sr/Mn threshold of 100 (Sørensen
et al., 2015). However, maintaining thresholds for diagenetic screening is relatively arbitrary and the height
of the thresholds used differs widely in the literature (e.g. Veizer, 1983; Steuber et al., 2002; Ullmann and
Korte, 2015; de Winter and Claeys, 2016). Applying these thresholds risks introducing biases to chemical
datasets from fossil shells and may not be an ideal method for diagenetic screening. Furthermore, large
variation in the *in vivo* incorporation of Mn and Sr in mollusk shell carbonate and a strong dependence on
the diagenetic setting can make the interpretation of shell preservation from trace element ratios alone
highly ambiguous (Ullmann and Korte, 2015). This conclusion is supported by the trace element and stable
isotope data gathered and compiled in this study (**Fig. 5**). Comparison of data from different fossil species
in Ivö Klack with two closely related modern oyster species from different environments indicates that the
differences between fossil mollusk species are similar to the differences among modern oyster species. It
also shows that pristine carbonate from modern oyster shells can contain up to 200 μg/g Mn accompanied
by a wide range in Sr concentrations.
One should be cautious when directly comparing trace element concentrations in biogenic calcite between
different time periods, as seawater composition of Late Cretaceous oceans (e.g. concentrations of Mg, Ca,
Sr and especially Li) may have been different from that of the present-day ocean (Stanley and Hardie, 1998;
Coggon et al., 2010; Rausch et al., 2013). For this reason, one would expect, for example, that Sr
concentrations in Late Cretaceous biogenic carbonate would be twice as low as those in carbonates formed
in the modern ocean (Stanley and Hardie, 1998; de Winter et al., 2017a). Trends in Mn and Sr
concentrations observed in all fossil species from Ivö Klack (**Fig. 5b**) likely point towards a diagenetic
process affecting a subset of the data. When observing variations in $\delta^{18}O$ (a sensitive proxy for diagenesis
and recrystallization; Brand and Veizer, 1980; Al-Aasm and Veizer, 1986b; **Fig. 5d**), the lack of covariation
between Mn concentration and $\delta^{18}O$ shows that there is little evidence for meteoric diagenesis in these
shells (Ullmann and Korte, 2015). Instead, these patterns are best explained by early marine cementation
of porous carbonate structures from sea water with similar temperature and $\delta^{18}O$ as the living environment
(see also Sørensen et al., 2015). These complex patterns merit great care in applying simple, general
thresholds for diagenesis. Therefore, in this study, a multi-proxy approach is applied for diagenetic
screening in which data is excluded based on a combination of Si, Ca, Mn, Fe and Sr concentrations, $\delta^{18}O$
values as well as SEM and visual observations of the shell structure at the location of measurement.
4.2 Dating of the Ivö Klack locality
Strontium isotope dating places the Ivö Klack deposits at 78.14 ± 0.26 Ma (**Fig. 4**). Nevertheless, age
estimates from strontium isotope analyses could be biased towards a younger age due to the influx of
radiogenic strontium-rich weathering products from the nearby Transscandinavian Igneous Belt (Högdal et
al., 2004). This may explain the fact that, when plotting the obtained age of 78.14 Ma on the compilation by
Wendler (2013), the age of the Ivö Klack falls slightly above the early/late Campanian subdivision (which is
placed at ~78.5 Ma), while the *B. mammilatus* biozone is defined as late early Campanian. However, studies
of modern strontium isotope ratio variability (Palmer and Edmond, 1989) and the potential bias of strontium
isotope ratios in shallow-water carbonates (Kuznetsov et al., 2012; Meknassi et al., 2018) show that the
effect of such inputs on strontium isotope dating results is generally negligible, except in semi-confined
shallow-marine basins characterized by considerable freshwater input and low salinities (<7 g/kg). No
evidence for such exceptional conditions at Ivö Klack exist (see **section 4.3**). We therefore conclude that
our strontium isotope age estimate, together with biostratigraphic constraints, places the Ivö Klack locality
in the latest early Campanian.
The refined dating of the Ivö Klack deposits and fossils allows the results of sclerochronological
investigations presented in this work to be placed in the context of longer-term climate reconstructions with
improved precision. While previous attempts at dating Campanian strata mainly focused on relative dating
using magneto- and biostratigraphy (Montgomery et al., 1998; Jarvis et al., 2002; Voigt et al., 2010),



integration of cyclostratigraphic approaches in this integrated stratigraphic framework has recently allowed
to constrain the age of the Campanian deposits more precisely (Voigt and Schönfield, 2010; Thibault et al.,
2012; Wendler, 2013; Thibault et al. 2016). Unfortunately, these attempts rarely cover the time interval in
which the Ivö Klack sediments were deposited (latest Early Campanian; e.g. Wendler, 2013; Perdiou et al.,
2016). Given the length of individual magnetochrons, carbon isotope shifts and biozones, the accuracy of
dating obtained by strontium isotope dating cannot, at the moment, be matched by the abovementioned
integrated stratigraphical approaches (Wagreich et al., 2012). For short, nearshore sections that cannot be
replaced within a long-term stratigraphic framework (such as Ivö Klack), strontium isotope stratigraphy on
well-preserved samples remains the most reliable dating method at present.
4.3 Ontogeny, metabolism and environment
The complex relationship between $\delta^{13}$C and $\delta^{18}$O records in *R. diluvianum* suggests that multiple factors
influence the incorporation of carbon into the shell calcite. In marine mollusks, dissolved inorganic carbon
(DIC) in the ambient sea water contributes to the majority (90%) of carbon used for shell mineralization
(McConnaughey, 2003; Gillikin et al., 2007). However, changes in respiration rates can alter the carbon
budget of shell carbonate by adding or removing isotopically-light respired carbon in the form of $CO_2$
(Lorrain et al., 2004). Of course, environmental changes in DIC can also have a strong influence on this
carbon budget, especially when bivalves grow in nearshore or estuarine conditions with large (seasonal)
variations in environmental $\delta^{13}$C of DIC and organic carbon (Gillikin et al., 2006). Conceptual models exist
that attempt to correlate shell $\delta^{13}$C in modern mollusks to environmental and physiological variations, but
these require knowledge of ambient $CO_2$ pressures and $\delta^{13}$C values of DIC, gas ventilation rates in the
animal and $CO_2$ and $O_2$ permeabilities of membranes (McConnaughey et al., 1997). Since these boundary
conditions are not available in fossil bivalve studies, the following discussion will remain limited to qualitative
interpretations of $\delta^{13}$C trends.
A part of the variation in $\delta^{13}$C may be explained by the presence of ontogenetic trends. These trends are
known to occur in marine and freshwater bivalves as well as in bivalves with symbionts (Klein et al., 1996b;
Watanabe et al., 2004; Gillikin et al., 2007; McConnaughey and Gillikin, 2008). The scale and direction of
these trends in $\delta^{13}$C are not consistent between individual *R. diluvianum* shells, which is also the case in
other bivalve species (see **section 3.5**; McConnaughey and Gillikin, 2008 and references therein). Studies
of modern bivalves show that in larger (older) bivalves, the contribution of respired $CO_2$ to carbon in the
shell is larger (up to 40%; Gillikin et al., 2007). This finding explains common trends of reducing $\delta^{13}$C with
age in bivalve shells, since respired carbon is isotopically lighter than environmental DIC. Since ontogenetic
trends are likely caused by changes in the amount of respired carbon entering the shell, and the direction
of these trends in *R. diluvianum*, the contribution of respired $CO_2$ to *R. diluvianum* shells likely did not strictly
increase with age. While this complicates the interpretation of $\delta^{13}$C records, the relative contribution of
environmental changes to $\delta^{13}$C variability in *R. diluvianum* shells does appear to be highest on the positive
end of the ontogenetic trend.
In all $\delta^{13}$C records we observe that the parts of the record that exceed a $\delta^{13}$C value of ±3.6‰ exhibit more
regular variations of ±0.6‰ that are correlated to the seasonal variability in $\delta^{18}$O (see **S6**). These periods
of covariation between $\delta^{13}$C and $\delta^{18}$O do not dominate in the records, as is evident from the lack of
seasonality in the annual stack of $\delta^{13}$C (**Fig. 8**). It is possible that, during parts of the lifetime of *R. diluvianum*
when the effect of respiration on $\delta^{13}$C of the shell is reduced, $\delta^{13}$C fluctuations reflect a combination of
changes in DIC and/or salinity in the environment, which are likely paced to the seasonal cycle. These
±0.6‰ shifts in $\delta^{13}$C that appear to be seasonal are much smaller than those in modern oyster records (2-
3‰ in low-latitude estuarine *Crassostrea virginica*; Surge et al., 2001; 2003; Surge and Lohmann, 2008).
Instead, the determined shifts more closely resemble the 0.5‰ variability in $\delta^{13}$C observed in modern
*Crassostrea gigas* from the same approximate latitude as Ivö Klack in the North Sea (Ullmann et al., 2013).
The extreme isotopic shifts in the estuarine *C. virginica* specimens have been shown to be caused by large
shifts in freshwater input due to large seasonal variations in rainfall over southern North America (Surge et
al., 2003), while smaller variations in *C. gigas* from the North Sea are produced by DIC changes due to
seasonal changes in productivity (e.g. spring blooms; Ullmann et al., 2013). The closer resemblance of *R.
diluvianum* to the North Sea condition evidences that the Ivö Klack paleoenvironment did not experience
large seasonal shifts in freshwater input and may have seen productivity peaks in spring season. The latter
interpretation is in agreement with the coincidence of negative $\delta^{13}$C excursions (in parts of the records not



affected by ontogenetic trends and respiration) with the low-$\delta^{18}$O season (winter or spring; **S6**) and the
occurrence of spawning and a peak in growth rates in the spring season (much like in wild modern oysters;
Berthelin et al., 2000; **Fig. 8,10a**). Large shifts in freshwater input are unlikely to have occurred in the Ivö
Klack setting, lending more confidence to the growth and temperature modelling based on $\delta^{18}$O records,
which requires the assumption that changes in $\delta^{18}$O$_{seawater}$ did not exert dominant control on the $\delta^{18}$O in
shell carbonate.
4.4 Temperature seasonality
Modelling of seasonal changes in growth rate and temperature based on the $\delta^{18}$O records in *R. diluvianum*
yielded a MAT of 18.7°C with an average seasonal range of 5.2°C (**Fig. 8**). The reconstructed MAT is 7-8
degrees warmer than the present-day 10-12°C mean annual sea surface temperature in the North and
Baltic seas at the same latitude (50-55°N; IRI/LDEO Climate Data Library, 2018). The MAT found in this
study is similar to the MAT of the late early Campanian Boreal Chalk Sea waters of 17-19°C based on long-
term reconstructions (Lowenstam and Epstein, 1954; Jenkyns et al., 2004; Friedrich et al., 2005; Thibault
et al., 2016) and is slightly warmer than mean annual air temperatures reconstructed at the same
paleolatitude (±15°C; Amiot et al., 2004). Averaging seasonality (**Fig. 8**) underestimates the extent of
seasonality at Ivö Klack, because not all seasons contributing to the average have long growing seasons,
which will reduce the average extent of seasonality. A more accurate estimate of the seasonal extent is
obtained by calculating the seasonal range from the coolest winter temperatures (12.6°C in *R. diluvianum*
4; **SI10**) with the warmest recorded summer temperature (26°C in *R. diluvianum* 1; **S10** which yields a
maximum seasonal sea surface temperature range of ±13.4°C. This is significantly less than the 16-20°C
temperature seasonality that occurs in the present-day Baltic and North seas at the same latitude as Ivö
Klack (IRI/LDEO Climate Data Library, 2018). Data on temperature seasonality in the Late Cretaceous is
scarce, especially in high-latitude settings. However, comparison with data on Cretaceous seasonality
between 15°N and 35°N paleolatitude (Steuber et al., 2005) shows that while MAT at 50°N was significantly
lower than those at lower latitudes (18°C vs. 25-30°C respectively), the seasonal temperature range during
cooler periods in the Late Cretaceous was remarkably similar between latitudes (10-15°C in subtropical
latitudes vs. ±14°C in this study). This observation contrasts with the present-day situation in Northern
Africa and Europe, in which seasonal temperature ranges are generally much higher in mid- to high-
latitudes (30-50°N) than in lower latitudes (10-30°N; Prandle and Lane, 1995; Rayner, 2003; Locarnini et
al., 2013; NOAA, 2018). Such seasonalities reconstructed from bivalve shells are not consistent with model
predictions of an ice-free Cretaceous world, since those models predict both smaller seasonal temperature
ranges and a shallower paleotemperature gradient (Barrera and Johnson, 1999; Hay and Floegel, 2012;
Upchurch et al., 2015).
4.5 Trace element proxies
4.5.1 Mg/Ca
From the data in **Fig. 8,** it is evident that there is a positive correlation between Mg/Ca and $\delta^{18}$O, or a
negative correlation between Mg/Ca and temperature. This correlation is opposite to the temperature-
relationships found in modern oyster species (Surge and Lohmann, 2008; Mouchi et al., 2013; Ullmann et
al., 2013). Furthermore, the difference between seasonally high and low Mg/Ca values is small (1.2
mmol/mol) compared to seasonal variability observed in modern oysters (4-10 mmol/mol; Surge and
Lohmann, 2008; Mouchi et al., 2013) and the variability between specimens of *R. diluvianum* (>3 mmol/mol;
**Fig. 7**). This dampening of the Mg/Ca cycle likely results from phase shifts between seasonal Mg/Ca cycles
in different specimens, causing seasonal cyclicity in different years and individuals to partly cancel each
other out in the annual stacks in **Fig. 8** (see **SI10**). These inconsistencies and the inverse temperature
correlation compared to modern oyster species demonstrate that it is unlikely that Mg/Ca ratios in *R.*
*diluvianum* are predominantly controlled by water temperatures. Mg/Ca ratios can therefore not be used as
reliable temperature proxies in this species.
4.5.2 Sr/Ca
Previous studies on modern bivalve species indicate that Sr/Ca ratios are not a likely candidate for
reconstructing temperature (Gillikin et al., 2005; Schöne et al., 2013; Ullmann et al., 2013). However, the
negative seasonal correlation between $\delta^{18}$O and Sr/Ca ratios (**Fig. 8**) suggests that there is at least some
seasonal parameter influencing Sr incorporation into *R. diluvianum* shells. This correlation cannot be



explained by classic diagenetic alteration of the shell, since this process would cause more negative $\delta^{18}O$
values to coincide with lower Sr concentrations (Brand and Veizer, 1980; Ullmann and Korte, 2015;
Sørensen et al., 2015), while the opposite is observed here. Unlike the Mg/Ca seasonality, comparison
between Sr/Ca variability in **Fig. 7** and **Fig. 8** shows that the seasonal variability in Sr/Ca is much less
dampened by inter-specimen variability and that phase relationships between Sr/Ca and $\delta^{18}O$ are
consistent between individuals (see also **S6**). The variability in Sr/Ca observed in foliate calcite in *R.*
*diluvianum* resembles seasonal variability in the same microstructure in modern *Crassostrea gigas* oysters
grown in a similar, though cooler, environment (see discussion in **section 4.3**) both in relation to the $\delta^{18}O$
cycle and in absolute Sr/Ca values (0.8-1.0 mmol/mol; Ullmann et al., 2013). This resemblance would
support a similar explanation for *R. diluvianum* as was attributed to Sr/Ca ratios in *C. gigas*, namely that
the proxy reflects seasonal changes in ambient sea water chemistry. There is some uncertainty as to
whether sea water Sr/Ca ratios in the Late Cretaceous were lower than (Stanley and Hardie, 1998; Coggon
et al., 2010) or similar to (Steuber and Veizer, 2002; Lear et al., 2003) those in the modern ocean. Local
enrichments in seawater Sr concentrations at Ivö Klack driving increased Sr composition in *R. diluvianum*
are unlikely, since Sr/Ca ratios exhibit only small (2-3%) lateral variability in the world's oceans (De Villiers,
1999). Therefore, the similarity in absolute calcite Sr/Ca ratios between modern *C. gigas* and Campanian
*R. diluvianum* may demonstrate that *R. diluvianum* incorporated more Sr into its shell than modern oysters
compensating for lower ambient Sr concentrations.
4.5.3 Li-proxies
While tentative temperature reconstructions based on Sr/Li and Mg/Li ratios (**Fig. 7**) appear consistent with
those found using $\delta^{18}O$, the stack in **Figure 8** shows that these ratios do not correlate with the seasonal
$\delta^{18}O$ cycle. Instead, it seems as if both Mg/Li and Sr/Li follow the same pattern with two maxima per annual
cycle. This, together with the strong covariation between Mg/Li and Sr/Li, is inconsistent with the
temperature dependence of these proxies (see **Fig. 7**). Instead, this covariation points to strong variations
in Li concentrations in the shells as drivers for the observed variability. The negative correlation between
Sr/Ca and Mg/Ca found in **Fig. 8** contradicts the inter-shell correlation between Mg/Li and Sr/Li found in
**Fig. 7**. This shows that, when comparing proxy records between shells, it is important to apply reliable age
models to correctly align the records such as the growth and age modelling approach applied in this study.
The age model-based approach reliably visualizes correlations between proxies on a seasonal scale, while
the approach of comparing seasonal averages and ranges of proxies (**Fig. 7**) puts more emphasis on
absolute inter-shell differences in the expression of proxies. While the latter may be useful in detecting
specimen-specific vital effects in trace element proxies (Freitas et al., 2008), the seasonally aligned
comparison in **Fig. 8** more reliably reveals relationships between proxies and can be used to infer
temperature dependence.
The inter-specimen comparison (**Fig. 7**) and the presence of randomly distributed ontogenetic trends in
Li/Ca (see **section 3.5**) suggests that a large part of the variability in Mg/Li and Sr/Li is controlled by
mechanisms that are local or even specimen-specific. The apparent occurrence of two peaks per year in
these records (**Fig. 8**) shows that sub-annual changes in environment may contribute to the variability in
Li-proxies in *R. diluvianum*. Riverine input can be a large source contributing to the dissolved Li budget in
shallow marine systems (Huh et al., 1998; Misra and Froelich, 2012). Therefore, synchronous fluctuations
in Mg/Li and Sr/Li ratios observed in **Fig. 8** may reflect changes in riverine input over the year. However,
stable isotope ratios in *R. diluvianum* show no sign of large fluctuations in freshwater input (see **section**
**4.3**), so the effect of these potential influxes on the local Li budget must have been limited. Furthermore,
dissolved Li in modern rivers strongly covaries with Mg and Sr, causing an increase in freshwater input to
have a limited effect on Mg/Li and Sr/Li ratios (Huh et al., 1998; Brunskill et al., 2003). The observation that
the inter-species variability in these proxies is much larger than the sub-annual variability (50-300 mol/mol
for Sr/Li and 350-1000 mol/mol for Mg/Li between specimens compared to 120-260 mol/mol for Sr/Li and
450-900 mol/mol for Mg/Li within a year) indicates that the effect of sub-annual environmental change is
likely to be small, and specimen-specific effects dominate. These complications prevent the use of Mg/Li
and Sr/Li proxies for temperature reconstructions in *R. diluvianum*.
The complexity of interpreting trace element proxies in this study shows that the incorporation of Mg and Li
into *R. diluvianum* was likely heavily biologically regulated. This result demonstrates that earlier successful
attempts to establish calibration curves for Li- and Mg-based temperature proxies (e.g. Füllenbach et al.,



2015; Dellinger et al., 2018) are probably strictly limited to bivalve species or close relatives. The same
conclusion was also drawn by Dellinger et al. (2018) based on Li/Mg and Li isotope ratio measurements in
biogenic carbonates. The lack of Mg/Li or Sr/Li calibrations in modern oyster shells limits the interpretation
of results in this study and establishing such calibrations using modern oysters in cultured experiments may
allow these proxies to be used for reconstructions from fossil oyster shells in the future.
4.6 Growth and life cycle
Modelling the growth of *R. diluvianum* shells based on $\delta^{18}O$ profiles (Judd et al., 2018) yields a lot of
information about the growth and life cycle of these oysters (**Fig. 9-10**). One of the most interesting results
is the remarkable similarity in growth patterns between individuals of *R. diluvianum* (**Fig. 9**). Except for the
final parts of growth curves of some of the older shells, all shells show similar development of shell height
with age. This development is well approximated by a Von Bertalanffy curve with a K value of 0.32 and a
theoretical full-grown shell height ($H_{max}$) of 120.3 mm (r = 0.89; ρ = 0.87; Von Bertalanffy, 1957; **Fig. 9**). The
consistency in growth curves between individuals of *R. diluvianum* is somewhat surprising given the fact
that modern oyster species are known to exhibit large variations in growth rates and shell shapes as a
function of their colonial lifestyle, which often limits the growth of their shells in several directions (Galtsoff,
1964; Palmer and Carriker, 1979). The strong resemblance of growth between individuals and the close fit
of the idealized Von Bertalanffy growth model suggests that growth of *R. diluvianum* at Ivö Klack was
relatively unrestricted in space. This hypothesis is consistent with the apparent mode of life of *R. diluvianum*
in Ivö Klack cemented together in groups, subject to strong wave action and turbulence, but with little
competition for space due to the high-energy environment (Surlyk and Christensen, 1974; Sørensen et al.,
2012). The shape of the growth curve of *R. diluvianum* is fairly consistent with that of modern Chesapeake
Bay oysters (*Crassostrea virginica*), which exhibit a slightly larger modelled maximum height (150 mm) and
a slightly smaller K-value (0.28). A larger subset of *R. diluvianum* specimens studied by Sørensen et al.
(2012) demonstrates that these bivalves could grow up to 160 mm in height. The curvature of the growth
of *R. diluvianum* (K -value) is also similar to that found for other modern shallow marine bivalve species
(e.g. *Macoma balthica*, K = 0.2-0.4; Bachelet, 1980; *Pinna nobilis*, K = 0.33-0.37; Richardson et al., 2004)
and significantly higher than in growth curves of deep marine bivalves (e.g. *Placopecten magellanicus*, K =
0.16-0.24; MacDonald and Thompson, 1985; Hart and Chute, 2009) or bivalves from cold habitats (e.g.
North Atlantic *Arctica islandica*, K = 0.06; Strahl et al., 2007). This reflects the high growth rates (steeper
growth curves, higher K-values) of shallow marine bivalves compared to species living in more unfavorable
or restricting (colder or deeper) habitats, with *R. diluvianum* clearly being part of the former group.
**Figure 10** and **Table 2** illustrate statistics of growth and seasonality for a total of 58 years of growth in the
complete dataset. This data indicates that the growing season is shorter than 365 days in all but five
modelled years, demonstrating that growth stops did occur in *R. diluvianum*. Minimum growth temperatures
(temperatures by which growth stops) are concentrated around 17°C ($\chi^2$ = 0.0088; **Fig. 10a**) and correlate
strongly to MAT (Pearson's r = 0.752; **Fig. 10b**), suggesting that while potential growth halts in *R.*
*diluvianum* occur systematically at a certain time interval of the year (first half of "winter"), they are not
forced by an absolute temperature threshold, but rather by timing relative to the seasonality (circadian
rhythm). On average, the moment of minimum growth occurs right after the highest temperatures of the
year are reached (early autumn, **Fig. 8**).
The spawning season (onset of the first growth year, see **3.9**) is concentrated in the two last months before
the $\delta^{18}O$ maximum (first half of "winter") when modelled water temperatures are ±17°C (**Fig. 10c**). Note that
only three of the five shells allowed sampling of the first month of growth, and extrapolated records for the
other two shells yielded spawning around the $\delta^{18}O$ minimum ("summer"). The offset of these estimates
likely results from uncertainty introduced due to extrapolation of the records of these two remaining shells,
showing that these estimates are likely unreliable. Comparing **Fig. 10c** and **Fig. 10a** shows that growth
halts and spawning occur at similar temperatures (16.85 ± 0.67°C and 16.98 ± 0.34°C respectively, p =
0.717), suggesting that these events occur simultaneously or on either side of a seasonal growth halt (if it
occurs).
**Figure 10c** shows that the distribution of months with fastest growth rate is random (p($\chi^2$) = 0.055, <95%
confidence). However, in 27 of the 58 years, the growth peak occurs in the season with decreasing $\delta^{18}O$
values ("spring season"), just after the moment of spawning (winter season; **Fig. 10a-b**). **Table 2** shows
that the extent of temperature seasonality (difference between minimum and maximum $\delta^{18}O$ converted to

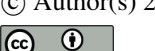



temperature) significantly influences the length of the growing season (strong correlation), the maximum
growth in that year and the total annual growth (weak correlations). MAT is a weak but significant driver of
annual growth, maximum growth and length of growing season. Ontogenetic age of the organism does not
predict a significant part of any of the above mentioned growth and seasonality parameters (**Table 2**). All
this suggests that temperature seasonality may not have been the dominant factor causing growth
cessations in *R. diluvianum*. This hypothesis is supported by the observation that temperatures at which
growth cessations occur (16.85 ± 0.67°C; **Fig. 10b**) show large variability and do not correspond
significantly to the lowest temperatures of the year.
This pattern is decidedly different from that observed in modern *Crassostrea gigas* shells, which generally
stop growing their shell at temperatures below ±10°C (Surge et al., 2001; Lartaud et al., 2010; Ullmann et
al., 2013). In contrast, lower latitude *Crassostrea virginica* from estuarine environments cease shell growth
at temperature maxima (>28°C; Surge et al., 2001). Other bivalves are known to have more flexible
temperature thresholds for shell precipitation (Ivany, 2012), but a lack of correlation between shell age and
length of season or minimum growth temperature (**Table 2**) demonstrates that there is no evidence for this
in *R. diluvianum*. These observations do not necessarily show that *R. diluvianum* tolerated larger
temperature differences than these modern taxa, because the maximum extent of seasonality between
12.6°C and 26°C reconstructed from $\delta^{18}$O records in this study (see **section 4.3**) causes neither the lower
nor the upper limit of temperature tolerance in modern oysters to be reached. If temperature tolerance of
*R. diluvianum* did resemble that of its closest modern relatives, then the mild seasonal temperature cycle
at Ivö Klack might have provided the ideal temperature conditions for its growth. Perhaps these favorable
conditions partly explain why biodiversity and abundance of invertebrates at Ivö Klack was so high (Surlyk
and Sørensen, 2010). If this was the case, then shell growth in *R. diluvianum* may not have been governed
by temperature, but rather by changes in productivity, as was already hypothesized based on fluctuations
in $\delta^{13}$C (see **section 4.4**). A strong 1:1 correlation between MAT and the temperature by which growth
cessations occur (slope = 0.981; r = 0.752; **Fig. 10c**) supports the hypothesis that absolute temperatures
did not limit shell growth, but rather that growth cessations occur consistently in certain parts of the seasonal
cycle. The observation that peak growth rates and spawning both occur during the early spring season
(**Fig. 10c**) is also consistent with the occurrence of spring blooms of increased productivity (**section 4.3**).
Finally, as **Table 2** shows, the length of the growing season positively correlates with the size of temperature
seasonality. This relationship is opposite to what would be expected if temperature controlled the growth of
*R. diluvianum* shells, since in that case, larger temperature seasonality would cause intolerable temperature
thresholds to be reached during larger parts of the seasonal cycle, which would shorten the length of the
growing season. Instead, the correlation in **Table 2** can be explained by a small input of isotopically light
freshwater in spring carrying nutrients to initiate the spring bloom (Arthur et al., 1983; Krantz et al., 1987).
Such a freshwater contribution would reduce $\delta^{18}$O$_{seawater}$ in the early spring season and dampen the
seasonality in shell $\delta^{18}$O values. A larger influence of seasonal freshwater input would cause longer growth
cessations to occur in the spring season, reducing the length of the growing season while also dampening
the reconstructed temperature seasonality, which explains the correlation found between these two
parameters. At the same time, this freshwater input would increase reconstructed MAT by increasing $\delta^{18}$O
values in *R. diluvianum* shells, explaining the weak positive correlation between MAT and length of the
growing season (**Table 2**). While seasonal changes in salinity and seawater $\delta^{18}$O must have remained
limited at Ivö Klack (see **section 4.3**), from the discussion above we conclude that seasonal differences in
productivity, potentially forced by input of nutrient-rich freshwater, are likely to have been a major factor
influencing shell growth in *R. diluvianum* at Ivö Klack. In this case, dampening of the seasonal $\delta^{18}$O cycle
may cause temperature seasonality reconstructions in this study to underestimate the real extent of
seasonality.

## 834 5. Conclusions

The highly biodiverse marine invertebrate community at Ivö Klack in the Kristianstad Basin in southern
Sweden offers a unique opportunity to recover a wealth of information about Campanian climate and
environment in high latitudes and the ecology and life of extinct invertebrate species that lived under these
conditions. The lack of burial and tectonic activity in the region favored *Rastellum diluvianum* fossil shells
from Ivö Klack to be well preserved, as is evident from the excellent preservation of growth structures typical



for ostreid shells as well as from limited evidence for geochemical changes associated with diagenetic
alteration. This excellent preservation allows the shells of *R. diluvianum* to be used to accurately and
precisely constrain the age of the Ivö Klack locality using strontium isotope stratigraphy (78.14 ± 0.26 Ma).
Furthermore, *R. diluvianum* shells reveal sub-annual scale variability in temperature, local environment and
growth rates through our multi-proxy geochemical approach. The combination of trace element and stable
isotope measurements with growth modelling based on $\delta^{18}O$ records in the shells allow all measured
proxies to be aligned on the same time axis. Application of transfer functions for potential Mg/Ca, Mg/Li and
Sr/Li temperature proxies established in modern invertebrates yields temperatures consistent with those
calculated from $\delta^{18}O$ records. However, close examination of the seasonal phase relationships between
these proxies reveals that the sub-annual variability in these trace element ratios is not controlled by
temperature changes alone. This observation supports previous studies that found the expression of trace
element proxies to be highly variable among species and even among different specimens of the same
species. If trace element proxies are to be used for seasonality reconstructions in pre-Quarternary times, a
more robust, non-species-specific model for the incorporation of trace elements by bivalves is required.
Establishing such a model requires culture experiments with different bivalve species in which multiple
parameters influencing trace element composition can be controlled (e.g. temperature, salinity, food intake
and microstructure).
Stable isotope records in *R. diluvianum* shells reveal a MAT of 17-19°C with a maximal seasonal water
temperature range of ±14°C (12.6°C - 26°C) at Ivö Klack. This value for MAT is consistent with long-term
temperature reconstructions in the Campanian Boreal Chalk Sea. Comparing the seasonal temperature
range reconstructed from *R. diluvianum* shells with other Late Cretaceous seasonality records from lower
latitudes reveals that temperature seasonality was remarkably similar across latitudes. These
reconstructions contradict results from climate models, which predict smaller temperature seasonalities.
This disagreement between data and models clearly illustrates the disadvantage of the lack of data on Late
Cretaceous seasonality outside the (sub-)tropical latitudes and highlights how important such
reconstructions are for improving our understanding of the dynamics in temperature variability in both space
and time during greenhouse climates.
Finally, the coupled modelling and multi-proxy approach applied in this study sheds light on the effects of
environmental changes on the life cycle and sub-annual growth of *R. diluvianum* shells. This study reveals
that growth curves of *R. diluvianum* strongly resemble those in modern shallow marine bivalves that grow
in coastal high latitude environments. However, changes in growth rate of our Boreal oysters seem
unrelated to temperature, in contrast to modern, high-latitude oysters that tend to lower their growth rate
and cease mineralization below a certain cold threshold. We conclude that growth cessations and sub-
annual changes in growth rate in *R. diluvianum* were most likely not caused by intolerable temperatures,
but rather by circadian rhythm tied to the seasonal cycle and seasonal changes in sea surface productivity,
driven by nutrient-rich freshwater inputs.

## Acknowledgements

This work was made possible with help of a IWT doctoral fellowship (IWT700) awarded to Niels de
Winter. Instrumentation at the VUB was funded by Hercules infrastructure grants (HERC9 and
HERC1309). The authors acknowledge financial and logistic support from the Flemish Research
Foundation (FWO, research project G017217N) and Teledyne CETAC Technologies
(Omaha, NE, USA) as well as support from VUB Strategic Research (BAS48). Stijn Goolaerts is funded
by a Belspo Brain project (BR/175/A2/CHICXULUB). The authors would like to thank David Verstraeten
for his help with stable isotope analyses. We thank Bart Lippens for assisting sample preparation and
Joke Belza for help with the LA-ICP-MS analyses. Thanks are due to Julien Cilis for his assistance with
SEM imaging. The authors wish to thank Emily Judd for discussions about her growth rate model for
bivalve shells and Roger Barlow for his assistance with combining strontium isotope measurements with
asymmetric error distributions.



**Supplementary files**

All supplementary files are stored in the open access online database Zenodo and can be accessed using the following link: **https://zenodo.org/record/2581305**

**S1**: High resolution (6400 dpi) scans of cross sections through the 12 shells of *Rastellum diluvianum* used in this study.

**S2**: Compilation of μXRF maps of cross sections through the 12 shells of *Rastellum diluvianum* used in this study.

**S3**: Compilation of XRF line scans measured through the foliated calcite of *Rastellum diluvianum* shells.

**S4**: Compilation of LA-ICP-MS data collected within the context of this study.

**S5**: Compilation of IRMS data used in this study.

**S6**: Composite figures of XRF linescan data through the shells of *Rastellum diluvianum*.

**S7**: Source code of the bivalve growth model adapted from Judd et al. (2018) including temperature equations for calcite.

**S8**: Compilation of strontium isotope data and ages used in this study.

**S9**: Compilation of the results from growth modelling on 5 *Rastellum diluvianum* shells.

**S10**: Compilation figures of proxy record data plotted on time axis for all 5 shells for which modelling was carried out.

**S11**: Plot of ontogenetic trends in $\delta^{13}$C and Li/Ca proxies including statistics on the spread of the slopes of these trends.

**S12**: Data on trends in $\delta^{13}$C and Li/Ca.

**S13**: Data used to create seasonality crossplots shown in **Fig. 7**.

**S14**: Data on statistics of the growth rates, seasonality and spawning season of all 5 bivalves for which modelling was done.

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
