# Peer review of "Shell chemistry of the Boreal Campanian bivalve Rastellum diluvianum (Linnaeus,"

_Biogeosciences, 2019_

## Referee Comment (RC1) · Johan Vellekoop (Referee) · 3 Feb 2020

The study of de Winter et al. presents interesting trace elemental and stable isotopic data from a set of Rastellum diluvianum specimens from the famous Campanian locality of Ivo Klack. The new datasets highlight both the potential of these kind of studies, and the complexity of interpreting trace elemental data. The authors have generated a wealth of data, providing valuable insights in the age of the Ivo Klack deposits (Sr isotopes), the local temperature seasonality (oxygen isotopes) and in the physiology of the studied oysters (carbon and oxygen istopes and elemental data). At the same

time, the complexity of the incorporation of trace elements in mollusk shells limits the usability of large parts of their data. The authors do a good job in highlighting this complexity, and show that, while sometimes elemental records of mollusk show cycling patterns, we are a long way away from successfully developing truly applicable proxies based on this time of data.

While the text is a bit lengthy, and some of the figures are rather complex, overall, this is a well-written manuscript. The authors have generated a substantial dataset, convincingly show that the studied specimens are well-preserved and provide interesting insights in the local climatic conditions at Ivo Klack. Their arguments are solid and their conclusions are sound.

Content wise, my only comments would be on the limited discussion on the possibility of a seasonal variability in d18O of seawater at Ivo Klack. They pass over this issue a bit too hastily, in my opinion. How is the assumption of a constant d18O of seawater justified? Wouldn't such a coastal site be susceptible for seasonal changes in riverine input? Particularly since the fennoscandian shield is usually placed in a wet/temperate climate belt, in Late Cretaceous climate reconstructions. The reference provided by the authors (Thibault et al., 2016) concerns a study on the chalks of the Stevns-1 core, which represents a much more distal site than Ivo Klack. Now, I realize that the authors are limited here, because constraining d18O of seawater is not easy, and I don't disagree with most of their general conclusions, but it would behoove them to acknowledge their uncertainties in this issue.

Apart from this, all my comments and suggestions are relatively minor. Therefore, I recommend this manuscript to be accepted with minor revisions.

Please also note the supplement to this comment:
https://www.biogeosciences-discuss.net/bg-2019-74/bg-2019-74-RC1-supplement.pdf

**Supplement:**

**Comments & suggestions:**

**P2, L54:** What does the "it" in this sentence exactly refer to? The Late Cretaceous cooling trend?

**P2, L57-59**: In the 90's chalk was still considered to record sea surface conditions faithfully. Over the last decades, this viewpoint has changed. Most chalk consists of recrystallized material. As a result, d18O values usually result in much lower temperatures, e.g. resulting in the (apparent) Cool Tropics Paradox. I advise the authors to read up on this. The values recorded by Jenkyns et al. are in all likelihood a large underestimation of SSTs (with even Cenomanian-Turonian values still below 28 degrees…). In reality, Cretaceous SST's were probably much higher. See for example the review paper of O'Brien et al., 2017.

**P2 L67-68**: With a Tethys ocean still present, a Panama corridor still present and closed-off Tasman and Drake Gateways, I wouldn't state that the continental configuration is "relatively modern". Yes: apart from India, most continents were already close to their present-day position, but from a climatological and paleoceanographical perspective, the Late Cretaceous continental configuration was widely different. Of course, this does not mean that the Campanian could be considered an interesting analogue. I just would not play the continental configuration card.

**P2 L73-75:** Does the data represent a fundamental component of the climate system? Or the seasonality? Please rephrase.

**P3 L93 "The incorporation of these chemical proxies into bivalve shells…":** This is a confusing sentence. Are the authors discussing the application of proxies on bivalve shells? Or are they concerned with the incorporation of chemical signals into bivalve shells?

**P3 L109-126 "The Kristianstad Basin….":** This paragraph feels a bit misplaced. There is a large jump from the previous paragraph (on the value of mollusks as archives of seasonality ) to this one (on the Kristianstad Basin). I think this paragraph would better fit directly after the first paragraph of the Introduction. The first paragraph of the section ends with the notion that Late Cretaceous seasonality records from high latitudes are scares. This could very easily be followed by "The Kristianstad Basin in Sweden provides a great potential for such a high latitude seasonality records. Particularly the Ivo Klack site, located on the southeastern Baltic…. Etcetera).

**P3 L110-112:** suggestion: "The coarsely latest early Campanian shallow marine sediments deposited at Ivö Klack consist of sandy and silty nearshore deposits containing carbonate gravel (Christensen, 1975; 1984; Surlyk and Sørensen, 2010; Sørensen et al., 2015)." (to avoid a confusing "and are.." construction.

**P3 L114:** maybe start a new sentence on the paleolatitude.

**P3 L115:** no glaciotectonic movements in this region?

**P3 L124:** I presume "original shell material " only refers to the calcitic material? Or is aragonite also preserved?

**P7 L195:** TSR and TSA are not specified. What do there abbreviations stand for?

**P8 L243-244:** what percentage of samples were run in duplicate?

**P11 L340-346:** There are a lot of 'allows' in this paragraph. Maybe rephrase a sentence or two?

**P11 L358-360:** This sentence is slightly confusing because the "(deeper waters)" directly follows the "bivalves". This reads as if the bivalves live in deeper waters, rather than the belemnites. Please rephrase.

**P13 L388-389:** How is the assumption of a constant d18O of seawater justified? Wouldn't such a coastal site be susceptible for seasonal changes in riverine input? Particularly since the fennoscandian shield is usually placed in a wet/temperate climate belt, in Late Cretaceous climate reconstructions. The reference provided by the authors (Thibault et al., 2016) concerns a study on the chalks of the Stevns-1 core, which represents a much more distal site than Ivo Klack.

**P13 L394-396 "Superimposed on these changes, a statistically significant ontogenetic trend can be discerned in the d13C records of 10 out of 12 shells. However, the scale and direction of these trends do not seem consistent between shells.":**

(1) I understood that only 5 of the 12 specimens were measured for isotopes? How can the authors have d13C data on all 12 shells? In table 1, only the 5 specimens are mentions, of which 3 out of 5 seem to have a statistically significant trend? It looks like something got mixed up here

(2) Please insert a reference to Table 1 here. This was not immediately clear from the text.

(3) I am intrigued by the difference in the direction of supposed ontogenetic trends. On the other hand, the only shell with a negative trend doesn't show a statistically significant trend….

**P13 L403:** Supplementary file S10 seems to be missing from the supplements

**P16 L451-453:** Is anything known about annual variations in growth rates in modern oysters? Do they respond to food availability? Could this be an early spring phytoplankton bloom? Or some other environmental parameter? Or is there a relationship with something like spawning?

**P20 L568:** salinities are usually not indicated in g/kg, but either in psu or in m%

**P21 L600:** "as well as" should be replaced by "including", since bivalves with symbionts are also marine or freshwater bivalves.

**P22 L644 "because not all seasons contributing to the average have long growing seasons":** seasons having long growing seasons? This is a confusing sentence. Please rephrase.

**P23 L696-698:** Why would oysters need to compensate for lower ambient Sr concentrations? What is the benefit of building Sr into their shells? How does this help to compensate for lower seawater Sr concentrations?

**P24 L774-782:** Is anything known about the spawning season of modern oysters? Maybe the authors could discuss how similar or dissimilar their results are.

**P25 L819-832:** The notion of a spring supply of freshwater, bringing in nutrients, causing a spring phytoplankton bloom, is somewhat conflicting with the assumption of a constant d18O of seawater, discussed in lines 388-389. Note to the authors: at modern day mid- to high latitudes, the spring bloom is often triggered by storm-induced mixing. A spring bloom is not necessarily related to riverine input of nutrients. It could be related to changes in mixed-layer depth as well..

---

## Referee Comment (RC2) · Andrew Johnson (Referee) · 4 Feb 2020

**Comments on de Winter et al. (submitted to Biogeosciences)**

This paper contains a great deal of carefully collected data but I think that it suffers from the sheer volume of information, and the attempt to discuss all issues to which the data may relate. Had the authors started with a question rather than with the data they would have developed a clearer line of argument, making the contribution easier to read, more persuasive and (I think, ultimately) more used. The main 'question' is probably seasonality in the Cretaceous, but we are led in various other directions, and certain important issues relating to the $\delta^{18}$O data go undiscussed in the process. By contrast, there is extensive discussion of the meaning of the trace-element information but these data in the end contribute nothing to the seasonality question – temperature variation is determined entirely from the $\delta^{18}$O data. There is a separate paper to be written on why the trace-element data does not help in determining seasonality. I suggest the authors focus here on doing a good job with the $\delta^{18}$O data (its implications for seasonality, together with those for growth) and deal only with trace-element data in so far as it relates to age and preservation.

With respect to the $\delta^{18}$O data my main query is the authors' abandonment of their initial estimate of seasonal temperature range (5.2°C) in favour of a much higher figure (13.4°C), representing the difference between the maximum and minimum temperatures from all the shells sampled. They then go on to compare this with figures for seasonal temperature range in the North Sea now and at lower latitudes in the Cretaceous, but it is not clear whether these figures are derived from equivalent (extreme) summer and winter values. If they are not the comparisons are worthless, and the conclusions about latitudinal seasonality variation in the Cretaceous compared to now will need to be reformulated. It looks like the figure for the North Sea now is based on extreme values (the stated range of 16–20°C is much higher than the mean range of about 11°C in the southern North Sea) but the authors need to explain this.

Another obscure use of the $\delta^{18}$O data is in Fig. 10. I looked at this, the caption, and the accompanying text for a long time but could not understand how the time of spawning was being inferred. The statement (LL 493–494) 'The onset of the first growth year in each shell at its precise position relative to the seasonal temperature cycle showed in which season spawning occurred (Fig. 10c)' does not mean anything to me – what is 'the first growth year'? The caption of part b added to my confusion since it does not describe what is illustrated—a bivariate plot of minimum growth temperature against mean annual temperature.

These two instances where further explanation is required of the use of $\delta^{18}$O data only emphasise the need to exclude discursive trace-element data and discussion, especially if (as recommended below) all the $\delta^{18}$O profiles are included in the main text.

Some other points:

LL54–55. How is the cooling trend 'recorded in the white chalk successions…'?

L99. The 'vital effects' largely relate to trace element content. A small effect on isotopic composition has been noted in *Pecten maximus* but little or no effect in other scallop species.

Fig. 3a. The use of the false yellow colour needs to be explained in the caption. What is the (non-sediment) yellow-coloured area – maybe altered pallial myostracum? If so, the early ontogenetic

samples would be from the inner shell layer – not ideal material (deposited far from the shell edge) and maybe an explanation for some aberrant data.

L 258. Some brief justification is required for the choice of value for water $\delta^{18}O$, even if it repeats Thibault et al. (2016) – this is an important issue in the present context.

L288. The parallelograms are not in 'different shades of blue'.

L348. Exclude 'multi-proxy' (redundant).

L368. Exclude 'vast' – there are quite a lot of $\delta^{18}O$ values associated with a Mn content of more than 100 μg/g.

L373. The results for *C. gigas* are not in 'grey/black'.

Fig. 6. Explain the vertical dashed lines (corresponding to the maxima in the $\delta^{18}O$ plot); change 1.0 to -1.0 for the water value on the y-axis. I think it would be worth having the $\delta^{18}O$ profiles from all the shells (not just this one) in the main text, so that the reader can get a picture of all the important data (see also comment on L457).

L425. 'virginica' in italics.

L437. 'follows' rather than 'shows'

LL450–451. You don't mean 'seasonal temperature range … was between 16°C and 21°C'. I suggest you say 'temperature varied between 16°C in winter and 21°C in summer'.

L457. This is where you need to be able to refer to all the $\delta^{18}O$ profiles.

Fig. 9. It is not clear to me how ages were derived for the start of the growth curves. Were growth increments used?

L583. 'placed' rather than 'replaced'.

LL664–665, 703–704. Repetitions of earlier statements.

LL752–3. 'cemented together in groups' suggests there would have been space competition and a 'high-energy environment' is not obviously something that would reduce space competition – needs explanation.

L760. 'deep shelf' for 'deep marine' – *Placopecten magellanicus* does not occur in anything other than shelf environments.

General point: please refer in the text to relevant parts of figures (where identified by letter) rather than the whole figure, to facilitate rapid appreciation of data.

---

## Author Comment (AC1) · 17 Feb 2020

Dear Aninda Mazumbar,

On behalf of my co-authors, I would like to thank dr. Johan Vellekoop and dr. Andrew Johnson for their insightful constructive comments on our manuscript. Attached, I will provide a point-by-point reply (in bold) to these comments (in italics) and state which changes we will make to the manuscript to take away the reviewers' concerns and prepare the text for publication. I hope that the suggested changes below will be

sufficient to allow us to revise our manuscript for publication in Biogeosciences, and I look forward to hearing from you concerning your decision.

Sincerely, Niels J. de Winter

Please also note the supplement to this comment:
https://www.biogeosciences-discuss.net/bg-2019-74/bg-2019-74-AC1-supplement.pdf

**Supplement:**

*Johan Vellekoop (Referee)*

*johan.vellekoop@kuleuven.be*

*The study of de Winter et al. presents interesting trace elemental and stable isotopic data from a set of Rastellum diluvianum specimens from the famous Campanian locality of Ivo Klack. The new datasets highlight both the potential of these kind of studies, and the complexity of interpreting trace elemental data. The authors have generated a wealth of data, providing valuable insights in the age of the Ivo Klack deposits (Sr isotopes), the local temperature seasonality (oxygen isotopes) and in the physiology of the studied oysters (carbon and oxygen istopes and elemental data). At the same time, the complexity of the incorporation of trace elements in mollusk shells limits the usability of large parts of their data. The authors do a good job in highlighting this complexity, and show that, while sometimes elemental records of mollusk show cycling patterns, we are a long way away from successfully developing truly applicable proxies based on this time of data.*

*While the text is a bit lengthy, and some of the figures are rather complex, overall, this is a well-written manuscript. The authors have generated a substantial dataset, convincingly show that the studied specimens are well-preserved and provide interesting insights in the local climatic conditions at Ivo Klack. Their arguments are solid and their conclusions are sound.*

*Content wise, my only comments would be on the limited discussion on the possibility of a seasonal variability in d18O of seawater at Ivo Klack. They pass over this issue a bit too hastily, in my opinion. How is the assumption of a constant d18O of seawater justified? Wouldn't such a coastal site be susceptible for seasonal changes in riverine input? Particularly since the fennoscandian shield is usually placed in a wet/temperate climate belt, in Late Cretaceous climate reconstructions. The reference provided by the authors (Thibault et al., 2016) concerns a study on the chalks of the Stevns-1 core, which represents a much more distal site than Ivo Klack. Now, I realize that the authors are limited here, because*

*constraining d18O of seawater is not easy, and I don't disagree with most of their general conclusions, but it would behoove them to acknowledge their uncertainties in this issue.*

**We acknowledge that the reconstruction of sea surface temperatures from stable oxygen isotopes suffers from assumptions about water oxygen isotope composition. We realize that we may not have given this fact the proper attention in our manuscript. In the revised version, we will therefore update our discussion of stable oxygen isotopes where these are translated to temperatures and make clear that these conversions are based on assumptions. We will add a paragraph at the beginning of the discussion of our stable oxygen isotope results in which we more clearly explain what assumptions we make about sea water composition. Finally, at the end our discussion of temperature seasonality we will discuss how the type of seasonal changes in sea water isotope composition that may be expected in a rocky shore setting may influence our conclusions.**

*Apart from this, all my comments and suggestions are relatively minor. Therefore, I recommend this manuscript to be accepted with minor revisions. Please also note the supplement to this comment:*

[https://www.biogeosciences-discuss.net/bg-2019-74/bg-2019-74-RC1-supplement.pdf](https://www.biogeosciences-discuss.net/bg-2019-74/bg-2019-74-RC1-supplement.pdf)

Comments in PDF supplement:

Comments & suggestions:

*P2, L54: What does the "it" in this sentence exactly refer to? The Late Cretaceous cooling trend?*

**Yes, this refers to the cooling trend, we will replace "It" by "The cooling trend" for clarity.**

*P2, L57-59: In the 90's chalk was still considered to record sea surface conditions faithfully. Over the last decades, this viewpoint has changed. Most chalk consists of recrystallized material. As a result, d18O values usually result in much lower temperatures, e.g. resulting in the (apparent) Cool Tropics Paradox. I advise the authors to read up on this. The values recorded by Jenkyns et al. are in all likelihood a large underestimation of SSTs (with even Cenomanian-Turonian values still below 28 degrees…). In reality, Cretaceous SST's were probably much higher. See for example the review paper of O'Brien et al., 2017.*

**This is a valid comment, and we will briefly discuss this later insight in our introduction. However, we do note that the introduction of previous work on chalk here mostly serves to introduce the reader into climate reconstructions from successions in the Boreal Chalk Sea. We would therefore rather add some discussion about the validity of SST reconstructions from such successions in the discussion section, where we compare different temperature estimates.**

*P2 L67-68: With a Tethys ocean still present, a Panama corridor still present and closed-off Tasman and Drake Gateways, I wouldn't state that the continental configuration is "relatively modern". Yes: apart from India, most continents were already close to their present-day position, but from a climatological and paleoceanographical perspective, the Late Cretaceous continental configuration was widely different. Of course, this does not mean that the Campanian could be considered an interesting analogue. I just would not play the continental configuration card.*

**Valid point, we will rephrase this and nuance our introduction of the Campanian as a reference for future climate, removing the notion of "relatively modern" continental configuration.**

*P2 L73-75: Does the data represent a fundamental component of the climate system? Or the seasonality? Please rephrase.*

**We will rephrase this to ", although seasonality constitutes a fundamental component of the climate system"**

*P3 L93 "The incorporation of these chemical proxies into bivalve shells…": This is a confusing sentence. Are the authors discussing the application of proxies on bivalve shells? Or are they concerned with the incorporation of chemical signals into bivalve shells?*

**Agreed, we will rephrase this sentence stating that the application of trace element proxies in bivalve shell records is complicated by vital effects.**

*P3 L109-126 "The Kristianstad Basin….": This paragraph feels a bit misplaced. There is a large jump from the previous paragraph (on the value of mollusks as archives of seasonality ) to this one (on the Kristianstad Basin). I think this paragraph would better fit directly after the first paragraph of the Introduction. The first paragraph of the section ends with the notion that Late Cretaceous seasonality records from high latitudes are scares. This could very easily be followed by "The Kristianstad Basin in Sweden provides a great potential for such a high latitude seasonality records. Particularly the Ivo Klack site, located on the southeastern Baltic…. Etcetera).*

**We thank the reviewer for this suggestion and indeed agree that this paragraph fits better straight after the introduction into the Boreal Chalk Sea reconstructions. We will move the paragraph to this location in the revised version and introduce bivalve shells as climate archives after the site description.**

*P3 L110-112: suggestion: "The coarsely latest early Campanian shallow marine sediments deposited at Ivö Klack consist of sandy and silty nearshore deposits containing carbonate gravel (Christensen, 1975; 1984; Surlyk and Sørensen, 2010; Sørensen et al., 2015)." (to avoid a confusing "and are.." construction.*

**We like this suggestion by the reviewer and will implement it with a minor change: "The coarsely uppermost lower Campanian shallow marine sediments deposited at Ivö Klack consist of sandy and silty nearshore deposits containing carbonate gravel (Christensen, 1975; 1984; Surlyk and Sørensen, 2010; Sørensen et al., 2015)."**

*P3 L114: maybe start a new sentence on the paleolatitude.*

**Agreed, we will rephrase this to "Late Cretaceous transgression. The paleolatitude of the site is 50°N."**

*P3 L115: no glaciotectonic movements in this region?*

**Post-glacial vertical crustal motion of the Kristianstad Basin is very limited (between -1 and +1 mm/yr), because the area is situated in the neutral uplift zone between compressed crust that is rebounding (most of the Scandinavian peninsula) and the glacial bulge (more to the south; Vestøl et al., 2019). The quiet tectonic history of this area is also documented in a report by Paulamäki & Kuivamäki (2006). Similar observations about the lack of glacio-eustatic rebound and other tectonic activity in the area have been documented by Surlyk and Sørensen (2010) and Christensen (1984).**

- **Christensen, W. K.: The Albian to Maastrichtian of southern Sweden and Bornholm, Denmark: a review, Cretaceous Research, 5(4), 313–327, 1984.**

- Paulamaeki, S. and Kuivamaeki, A.: Depositional history and tectonic regimes within and in the margins of the Fennoscandian shield during the last 1300 million years, Posiva Oy. [online] Available from: http://inis.iaea.org/Search/search.aspx?orig_q=RN:43061185 (Accessed 12 February 2020), 2006.
- Surlyk, F. and Sørensen, A. M.: An early Campanian rocky shore at Ivö Klack, southern Sweden, Cretaceous Research, 31(6), 567–576, 2010.
- Vestøl, O., Ågren, J., Steffen, H., Kierulf, H. and Tarasov, L.: NKG2016LU: a new land uplift model for Fennoscandia and the Baltic Region, J Geod, 93(9), 1759–1779, doi:10.1007/s00190-019-01280-8, 2019.

*P3 L124: I presume "original shell material " only refers to the calcitic material? Or is aragonite also preserved?*

**The oyster shells we describe contain very little original aragonitic shell structures (oysters only build thin aragonite structures at the resilium and the adductor muscle scar), so we only investigated calcite preservation in our study. The same holds true for the cited studies into macrofossils at this site. To clarify this, we will specifically refer to "calcite shell preservation" in the revised manuscript text.**

*P7 L195: TSR and TSA are not specified. What do there abbreviations stand for?*

**These stand for Time of Stable Accuracy and Time of Stable Reproducibility, terms which are defined in de Winter et al., 2017b. We will revise this section by writing out the full names of these terms and briefly defining what is meant by them in the context of microXRF measurement quality.**

*P8 L243-244: what percentage of samples were run in duplicate?*

**Duplicates were measured during every run of ~30 samples. We will mention this in the revised manuscript.**

*P11 L340-346: There are a lot of 'allows' in this paragraph. Maybe rephrase a sentence or two?*

**Good point, we rephrase the sentences on lines 341-346 to: "From this extrapolation we could estimate the total shell height from microstructural growth markers (Fig. 3; following Zimt et al., 2018), linking growth to changes in shell chemistry. This way, chemical changes in the shell can be interpreted in terms of environmental changes by applying calibration curves for trace element proxies that were previously established for modern oyster species (e.g. Surge and Lohmann, 2008; Ullmann et al., 2013; Mouchi et al., 2013; Dellinger et al., 2018)."**

*P11 L358-360: This sentence is slightly confusing because the "(deeper waters)" directly follows the "bivalves". This reads as if the bivalves live in deeper waters, rather than the belemnites. Please rephrase.*

**We rephrased this to: "This suggests that $\delta^{13}C$ in belemnite rostra are affected by vital effects while heavier $\delta^{18}O$ values of the belemnites suggest that belemnites lived most of their life away from the coastal environment (in deeper waters),"**

*P13 L388-389: How is the assumption of a constant d18O of seawater justified? Wouldn't such a coastal site be susceptible for seasonal changes in riverine input? Particularly since the fennoscandian shield is usually placed in a wet/temperate climate belt, in Late Cretaceous climate reconstructions. The reference*

*provided by the authors (Thibault et al., 2016) concerns a study on the chalks of the Stevns-1 core, which represents a much more distal site than Ivo Klack.*

**This comment reflects the major criticism of the reviewer. We hope that by more thoroughly discussing the stable oxygen isotope composition of sea water we can acknowledge the shortcomings of this assumption of constant seawater δ¹⁸O values.**

*P13 L394-396 "Superimposed on these changes, a statistically significant ontogenetic trend can be discerned in the d13C records of 10 out of 12 shells. However, the scale and direction of these trends do not seem consistent between shells.":*

*(1) I understood that only 5 of the 12 specimens were measured for isotopes? How can the authors have d13C data on all 12 shells? In table 1, only the 5 specimens are mentions, of which 3 out of 5 seem to have a statistically significant trend? It looks like something got mixed up here*

*(2) Please insert a reference to Table 1 here. This was not immediately clear from the text.*

*(3) I am intrigued by the difference in the direction of supposed ontogenetic trends. On the other hand, the only shell with a negative trend doesn't show a statistically significant trend….*

**We fully agree with all the reviewer's points of critique here, something must have gotten mixed up here and we apologize for the mistake. We will rephrase this sentence as follows: "Superimposed on these changes, a statistically significant ontogenetic trend can be discerned in the d13C records of 3 out of 5 shells. In specimens that show a statistically significant ontogenetic trend δ¹³C increases with age (see Table 1)".**

*P13 L403: Supplementary file S10 seems to be missing from the supplements*

**File S10 contains the plots of multiproxy records against age. We regret that these plots did not make it into the supplement and will make sure that they do in the revised version. In response to comments by the second reviewer, we now show δ¹⁸O records of all shells in the main manuscript as well.**

*P16 L451-453: Is anything known about annual variations in growth rates in modern oysters? Do they respond to food availability? Could this be an early spring phytoplankton bloom? Or some other environmental parameter? Or is there a relationship with something like spawning?*

**In the revised manuscript, we will add some discussion here about how these findings compare with those in modern oysters. There is some literature on this which suggests indeed that food availability plays a role. We hypothesize the presence of a spring phytoplankton bloom later in the manuscript, but will move this hypothesis forward here, where we can discuss it together with the comparison with modern oyster species.**

*P20 L568: salinities are usually not indicated in g/kg, but either in psu or in m%*

**We will convert these values to psu.**

*P21 L600: "as well as" should be replaced by "including", since bivalves with symbionts are also marine or freshwater bivalves.*

**Correct, we will rephrase this.**

*P22 L644 "because not all seasons contributing to the average have long growing seasons": seasons having long growing seasons? This is a confusing sentence. Please rephrase.*

**We agree that this is a convoluted sentence and will rephrase as follows: "Averaging seasonality (Fig. 8) underestimates the extent of seasonality at Ivö Klack, because not all specimens contributing to the average have long growing seasons, which will reduce the average extent of seasonality."**

*P23 L696-698: Why would oysters need to compensate for lower ambient Sr concentrations? What is the benefit of building Sr into their shells? How does this help to compensate for lower seawater Sr concentrations?*

**We agree that "compensate" is not the right term here. We will rephrase as follows: "Therefore, the similarity in absolute calcite Sr/Ca ratios between modern *C. gigas* and Campanian *R. diluvianum* demonstrates that *R. diluvianum* incorporated more Sr into its shell relative to the ambient seawater concentration. This observation may entail that there is a minimum Sr concentration that is favorable for oysters to incorporate, or that there is a fixed physiological limit to oyster's discrimination against building Sr into their shells that is independent of ambient Sr concentrations."**

*P24 L774-782: Is anything known about the spawning season of modern oysters? Maybe the authors could discuss how similar or dissimilar their results are.*

**Modern oysters typically spawn at the end of the spring season and spat settles in during summer. This makes our result for *R. diluvianum* different from the modern situation. We will acknowledge this in the revised manuscript and provide references for spawning of modern *C. gigas*.**

*P25 L819-832: The notion of a spring supply of freshwater, bringing in nutrients, causing a spring phytoplankton bloom, is somewhat conflicting with the assumption of a constant d18O of seawater, discussed in lines 388-389. Note to the authors: at modern day mid- to high latitudes, the spring bloom is often triggered by storm-induced mixing. A spring bloom is not necessarily related to riverine input of nutrients. It could be related to changes in mixed-layer depth as well..*

**We thank the reviewer for this comment and advice and will add this to the discussion. As mentioned in our reply to his major comment, we will discuss potential changes in seawater composition in more detail in the revised manuscript and specifically add a paragraph detailing how changes in seawater composition can affect our interpretation in terms of temperature seasonality.**
*This paper contains a great deal of carefully collected data but I think that it suffers from the sheer volume of information, and the attempt to discuss all issues to which the data may relate. Had the authors started with a question rather than with the data they would have developed a clearer line of argument, making the contribution easier to read, more persuasive and (I think, ultimately) more used. The main 'question' is probably seasonality in the Cretaceous, but we are led in various other directions, and certain important issues relating to the δ18O data go undiscussed in the process. By contrast, there is extensive discussion of the meaning of the trace-element information but these data in the end contribute nothing to the seasonality question – temperature variation is determined entirely from the δ18O data. There is a separate paper to be written on why the trace-element data does not help in determining seasonality. I suggest the authors focus here on doing a good job with the δ18O data (its implications for seasonality, together with those for growth) and deal only with trace-element data in so far as it relates to age and preservation.*

**This is a valid point, and agree that our trace element data does, in the end, not contribute as much to the seasonality story as we would have hoped initially. We would therefore largely follow the reviewer's suggestion and strongly limit our discussion of the trace element data. However, we do not fully agree that the trace element data by itself would stand alone in a manuscript. Therefore, we would like to keep discussing (albeit more briefly) the patterns in trace element concentrations we find in our specimens. Moreover, we believe that the comment raised here is also partly a result of the (admittedly somewhat chaotic) structure our manuscript inherited in our attempt to tie together several lines of evidence and reasoning about the species' paleobiology and living environment. Besides shortening the discussion of trace element results, we will also make an attempt to streamline the manuscript as a whole to make these lines of reasoning easier to follow.**

*With respect to the δ18O data my main query is the authors' abandonment of their initial estimate of seasonal temperature range (5.2°C) in favour of a much higher figure (13.4°C), representing the difference between the maximum and minimum temperatures from all the shells sampled. They then go on to compare this with figures for seasonal temperature range in the North Sea now and at lower latitudes in the Cretaceous, but it is not clear whether these figures are derived from equivalent (extreme) summer and winter values. If they are not the comparisons are worthless, and the conclusions about latitudinal seasonality variation in the Cretaceous compared to now will need to be reformulated. It looks like the figure for the North Sea now is based on extreme values (the stated range of 16–20°C is much higher than the mean range of about 11°C in the southern North Sea) but the authors need to explain this.*

**This is a valid comment, and we will reevaluate this part of the manuscript where we compare our seasonality results with modern and reconstructed seasonality data. Data such as SST profiles of the present-day North Sea will invariably show differences depending on where these data were sampled from (e.g. from which water depth or locality). We will therefore be more careful in stating how**

exactly the data were sourced, whether these are extreme seasonal ranges or (more conventionally) differences between extreme monthly temperatures and how they compare to reconstructed seasonal amplitudes.

*Another obscure use of the δ18O data is in Fig. 10. I looked at this, the caption, and the accompanying text for a long time but could not understand how the time of spawning was being inferred. The statement (LL 493–494) 'The onset of the first growth year in each shell at its precise position relative to the seasonal temperature cycle showed in which season spawning occurred (Fig. 10c)' does not mean anything to me – what is 'the first growth year'? The caption of part b added to my confusion since it does not describe what is illustrated—a bivariate plot of minimum growth temperature against mean annual temperature.*

**We acknowledge that Fig. 10 may not be very clear, and we will attempt to revise this figure to clarify what we would like to show here. The time of spawning could be placed relative to the seasonal stable oxygen isotope cycle by noting during which part of the annual cycle shell growth started. Assuming the regular variations in stable oxygen isotope composition reflect temperature seasonality, the season in which the bivalve started growing can be inferred from phase of the oxygen isotope sinusoid during onset of growth. We will clarify how this is achieved in more detail in the revised version of the manuscript.**

*These two instances where further explanation is required of the use of δ18O data only emphasise the need to exclude discursive trace-element data and discussion, especially if (as recommended below) all the δ18O profiles are included in the main text.*

*Some other points:*

*LL54–55. How is the cooling trend 'recorded in the white chalk successions…'?*

**The cited references are of studies in which (mostly) oxygen isotope records from these chalks have been used to document this cooling trend. For clarity, we will rephrase this sentence as: "The cooling trend is well documented in stable oxygen isotope records from the white chalk successions of the Chalk Sea, which covered large portions of northwestern Europe during the Late Cretaceous Period…"**

*L99. The 'vital effects' largely relate to trace element content. A small effect on isotopic composition has been noted in Pecten maximus but little or no effect in other scallop species.*

**Agreed, we will clarify this in the revised version. "Vital effects" on oxygen isotope composition in bivalves are rare, and most of them are thought to precipitate at or close to isotopic equilibrium.**

*Fig. 3a. The use of the false yellow colour needs to be explained in the caption. What is the (non-sediment) yellow-coloured area – maybe altered pallial myostracum? If so, the early ontogenetic samples would be from the inner shell layer – not ideal material (deposited far from the shell edge) and maybe an explanation for some aberrant data.*

**We will add a description of the yellow color in the figure caption. This is indeed the color we use to highlight highly altered shell material and sediment infilling, as seen in the XRF maps below.**

*L 258. Some brief justification is required for the choice of value for water δ18O, even if it repeats Thibault et al. (2016) – this is an important issue in the present context.*

This comment touches on the major comment posed by our other reviewer (dr. Johan Vellekoop). We hope that the changes we will make in reply to his comment will satisfy this comment as well.

*L288. The parallelograms are not in 'different shades of blue'.*

**Correct, this is a remnant of an earlier version of this figure. We will correct this by stating that the specimens are represented by parallelograms of different colors matching the probability distributions below.**

*L348. Exclude 'multi-proxy' (redundant).*

**Agreed, this will be removed.**

*L368. Exclude 'vast' – there are quite a lot of δ18O values associated with a Mn content of more than 100 µg/g.*

**Agreed, we will remove "vast"**

*L373. The results for C. gigas are not in 'grey/black'.*

**Correct, this again refers to a previous version of the figure. We have overlooked this error and will correct it in the revised version, stating that the results of *C. gigas* are in yellow/brown.**

*Fig. 6. Explain the vertical dashed lines (corresponding to the maxima in the δ18O plot); change 1.0 to -1.0 for the water value on the y-axis. I think it would be worth having the d18O profiles from all the shells (not just this one) in the main text, so that the reader can get a picture of all the important data (see also comment on L457).*

**We will add a sentence to the caption stating that the vertical dashed lines separate growth years. In addition, we will correct the typographic error in our assumed δ$^{18}$O$_{sw}$ value. We will add a composite figure displaying all δ$^{18}$O data used in this study.**

*L425. 'virginica' in italics.*

**Certainly, we will change this in the revised text.**

*L437. 'follows' rather than 'shows'*

**Correct, this will be rephrased.**

*LL450–451. You don't mean 'seasonal temperature range … was between 16°C and 21°C'. I suggest you say 'temperature varied between 16°C in winter and 21°C in summer'.*

**Agreed, we will rephrase this accordingly.**

*L457. This is where you need to be able to refer to all the δ18O profiles.*

**Agreed, we will refer to the composite figure we will add showing all δ$^{18}$O records here.**

*Fig. 9. It is not clear to me how ages were derived for the start of the growth curves. Were growth increments used?*

For most specimens, δ¹⁸O measurements were possible until very close to the onset of mineralization. For the specimens were this was not the case, we used a combination of annual growth increments and extrapolation of the $\delta^{18}O$-based age model to infer the age of the ontogenetically oldest $\delta^{18}O$ measurements. We will clarify this in the revised version.

*L583. 'placed' rather than 'replaced'.*

**Agreed, we will rephrase this.**

*LL664–665, 703–704. Repetitions of earlier statements.*

**Agreed, we will significantly shorten these sections about trace element concentrations and remove these repetitions. This is also in response to the major comments by the reviewer stating (rightfully) that the discussion of trace element patters distracts from the main seasonality discussion in the manuscript which is mostly based on $\delta^{18}O$ records.**

*LL752–3. 'cemented together in groups' suggests there would have been space competition and a 'high-energy environment' is not obviously something that would reduce space competition – needs explanation.*

**Here we wanted to refer specifically to the competition with other taxa, which would not thrive in this high-energy environment. In addition, the *in situ* distribution of oysters on the fossil rocky shored of Ivö Klack as documented in Surlyk and Christensen (1974) and Sørensen et al. (2012) shows that there is limited competition for space. We will rephrase "cemented together in groups" into "cemented next to each other in groups" to clarify that the oysters are not cemented on top of each other (as modern *C. gigas* often is) and have less space limitations that modern oysters.**

*L760. 'deep shelf' for 'deep marine' – Placopecten magellanicus does not occur in anything other than shelf environments.*

**Correct, we will change this throughout the text.**

*General point: please refer in the text to relevant parts of figures (where identified by letter) rather than the whole figure, to facilitate rapid appreciation of data.*

**In the revised manuscript, we will go through all the figure references and specify the parts of figures wherever relevant.**

---

## Referee Report (RR1)

**Shell chemistry of the Boreal Campanian bivalve *Rastellum diluvianum* (Linnaeus, 1767) reveals temperature seasonality, growth rates and life cycle of an extinct Cretaceous oyster.**

Niels J. de Winter[1], Clemens V. Ullmann[2], Anne M. Sørensen[3], Nicolas Thibault[4], Steven Goderis[1], Stijn J.M. Van Malderen[5], Christophe Snoeck[1,6], Stijn Goolaerts[7], Frank Vanhaecke[5], Philippe Claeys[1]

[1]AMGC research group, Vrije Universiteit Brussel, Pleinlaan 2, B-1050 Brussels, Belgium.

[2]Camborne School of Mines, University of Exeter, Penryn, Cornwall, TR10 9FE, UK.

[3]Trap Danmark, Agem All 13, DK-2970, Hørsholm, Denmark.

[4] Department of Geoscience and Natural Resource Management, University of Copenhagen, Øster Voldgade 10, DK-1350 Copenhagen C., Denmark.

[5]A&MS research unit, Ghent University Campus Sterre, Krijgslaan 281, Building S12, B-9000 Ghent, Belgium.

[6]G-Time laboratory, Université Libre de Bruxelles, 50 Avenue F.D. Roosevelt, B-1050, Brussels, Belgium.

[7]Directorate of Earth and History of Life, Royal Belgian Institute of Natural Sciences, Vautierstraat 29, B-1000 Brussels, Belgium.

*Correspondence to: Niels J. de Winter (niels.de.winter@vub.be)*

**Abstract**

The Campanian age (Late Cretaceous) is characterized by a warm greenhouse climate with limited land ice volume. This makes this period an ideal target for studying climate dynamics during greenhouse periods, which are essential for predictions of future climate change due to anthropogenic greenhouse gas emissions. Well-preserved fossil shells from the Campanian (±78 Ma) high mid-latitude (50°N) coastal faunas of the Kristianstad Basin (southern Sweden) offer a unique snapshot of short-term climate and environmental variability, which complement existing long-term climate reconstructions. In this study, we apply a combination of high-resolution spatially resolved trace element analyses (µXRF and LA-ICP-MS), stable isotope analyses (IRMS) and growth modelling to study short-term (seasonal) variations recorded in the oyster species *Rastellum diluvianum* from the Ivö Klack locality. Geochemical records through 12 specimens shed light on the influence of specimen-specific and ontogenetic effects on the expression of seasonal variations in shell chemistry and allows disentangling vital effects from environmental influences in an effort to refine palaeoseasonality reconstructions of Late Cretaceous greenhouse climates. Growth models based on stable oxygen isotope records yield information on the mode of life, circadian rhythm and reproductive cycle of these extinct oysters. This multi-proxy study reveals that mean annual temperatures in the Campanian higher mid-latitudes were 17 to 19°C with winter minima of ~13°C and summer maxima of 26°C, assuming a Late Cretaceous seawater oxygen isotope composition of -1‰VSMOW. These results yield Campanian latitudinal temperature gradients similar to the present, but with smaller latitudinal differences in temperature seasonality, 
[revised manuscript text omitted]

---

## Author Response (AR2)

Dear dr. Aninda Mazumdar and dr. Andrew Johnson,

On behalf of all authors, I would like to thank you for your additional comments on our
manuscript. We are glad to receive a positive reply to our revisions of the manuscript and are
happy to implement the additional minor revisions to the text to ready our manuscript for
publication. Please find our point-by-point rebuttal (in red) to these suggestions below, including
details on how we implement the necessary changes to the text, where applicable.

*Associate Editor Decision: Publish subject to minor revisions (review by editor) (14 Apr 2020) by*
*Aninda Mazumdar*

*Dear Authors,*

*Kindly go through the comments from the reviewer and submit the revised manuscript for a final*
*decision. You have made substantial improvement in the text, however some clarity is required*
*at few places as mentioned by the reviewer.*

*sincerely*

*Aninda Mazumdar*

Thank you for your moderation of the review process and for giving us the opportunity to
respond to these and previous review comments. We are looking forward to hearing your
decision.

*Suggestions for revision Referee #4 (dr. Andrew Johnson)*

*The authors have responded very positively to comments: the paper now has a clear 'thread'*
*and is therefore much easier to read and appreciate. The results and conclusions are novel and*
*important, and certainly merit publication. I have made a number of corrections and comments*
*(in square brackets) in the annotated version of the manuscript supplied alongside. Most will be*
*straightforward to address but the authors need to give a little more time to sections 5.5.2 and*
*5.5.3. Although I finally worked out what the authors were saying about spawning in section*
*5.5.2 (I had misunderstood their application of the term as spawning BY the individuals*
*investigated), some clarification of the text to prevent similar misunderstanding by other readers,*
*together with reconsideration/correction of other points in this and the next section (5.5.3), are*
*needed. Once these issues (and additional minor points) have been addressed the paper will be*
*ready for publication, and it will be an excellent contribution to the literature.*

We are glad to hear that dr. Johnson considers the manuscript improved, and are very happy
with his feedback that helped us to better structure our manuscript and clarify our discussion.
We regret that our discussion in section 5.5.2 was not clear and that we did not realize the
source of this confusion in our previous round of revisions. We will follow dr. Johnson's suggestions to prevent the same confusion from occurring with our future readers. We kindly
refer to the point by point rebuttal below regarding the other minor points raised by the reviewer.

**Minor in-text comments**

Line 29: "complement" changed to "complements"

Line 34: "allows" changed to "allow"

Line 40-41: These statements do not contrast

Correct, we rephrase to clarify that the "latitudinal gradients similar to the present" contrast with
the notion of "equable climate"

Line 68: "land ice free" rephrased to "essentially land-ice free"

Line 76: "risk being seasonally biased" rephrased to "risk seasonal bias"

Line 131: "these" was deleted

Line 163: Line break was inserted after "2.5 Aim"

Line 198: "on" rephrased to "at"

Line 210: The yellow area in the centre of the image is within the valve. We replace "between"
with "within"

Line 218: "Germany)." Dot added to sentence

Line 228: 'Regions of Interest' Quotation marks were added

Line 230: "selection" rephrased to "formulation"

Line 236: "as" rephrased to "so as"

Line 280: "sampled" rephrased to "extracted"

Line 324: "were" was added between "samples" and "placed"

Line 395: "as it does in" rephrased to "to"

Line 396: "From this extrapolation" rephrased to "On this basis"

Line 396: "…shell height from microstructural…" rephrased to "…shell height at specific times in
ontogeny from microstructural…"

Line 399: "analyses" was removed

Line 409: "diagenetic" rephrased to "diagenetically altered"

Line 434: "are" rephrased to "were"

Line 435: "5point" rephrased to "5-point"

Line 442: Use one scale of comparison - lower/higher or lighter/heavier. Strictly speaking a
value can't be light/heavy so low/high is better.

Agreed, we now use "lower" and "higher"

Line 474: "predictable" rephrased to "widely-shared"

Line 507: "variety" rephrased to "variation"

Line 521: "in this integrated stratigraphic framework" was deleted

Line 525-528: Repetition of material in section 4.1 - try at least to minimise. Also, replace
'subdivision' with 'boundary'.

Agreed, we shortened this section to "Strontium isotope dating places the Ivö Klack deposits at
78.14 ± 0.26 Ma (Fig. 4), slightly above the early/late Campanian boundary. When plotting the
obtained age of 78.14 Ma on the compilation by Wendler (2013), the age of the Ivö Klack falls
slightly above the early/late Campanian subdivision (which is placed at ~78.5 Ma), while the B.
mammilatus biozone is defined as late early Campanian (Wendler, 2013)."

Line 566: "the results of this stud." Rephrased to "our results for these element ratios"

Line 569: "likely" rephrased to "promising"

Line 572: "as basis" rephrased to "as a basis"

Line 580: "composition" rephrased to "content"

Line 582: "twice as low as in" rephrased to "half those of"

Line 583: "for example" was deleted

Line 583-584: "twice as low as" rephrased to "half"

Line 585: "remains" rephrased to "has remained"

Line 587: "entail" rephrased to "mean"

Line 587: "…fixed physiological limit to oyster's discrimination against building Sr into their
shells …" rephrased to "…physiologically fixed concentration of Sr in an oyster's shell…"

Line 587: "building" rephrased to "incorporating"

Line 588: If a similar amount of Sr is incorporated when the ambient concentration is lower,
doesn't it mean that the organism is still discriminating? Clarification needed.

Corrent, we rephrased this sentence clarifying that we hypothesize a fixed concentration of Sr in
oyster shells which is independent of the ambient seawater concentration (see comment on line
587).

Line 604: "(Fig. 6)" was added

Line 619: "values can be considerably lighter" rephrased to "values can be considerably lower"

Line 624: "…analyses (which does…)" Both either singular or plural.

Correct, this was rephrased to "…analyses (which do…"

Line 625: "which" was deleted

Line 635: "based on" rephrased to "from"

Line 638: "paleolatitude" was deleted

Line 644: for Ivö Klack material. Not necessarily from this site, we rephrase to "independent
marine temperature proxies" to clarify

Line 652: "such" was deleted

Line 653: "In addition" rephrased to "In this respect it is important to recognise that"

Line 657-658: Better expressed as winter minimum and summer maximum temperatures for
clarity, with the median added for comparison with Ivö Klack MAT (almost the same parameter).
Omit mention of mean annual SST because you don't actually supply it.

Agreed, we rephrased this sentence referring to "seasonal SST range" rather than MAT.

Line 660: maximum seasonal range in temperature [+ cite source].

This was rephrased accordingly. The source is the same as in the previous sentence.

Line 662: "seasonality rephrased to "temperature seasonality"

Line 663: "those" was deleted

Line 663: "vs." rephrased to "and"

Line 663: "C respectively" added "," between "C" and "respectively"

Line 671: "2015).5.5 Shell growth and ontogeny" added line break before "5.5"

Line 679: "Von" rephrased to "von"

Line 686: "Von" rephrased to "von"

Line 687: Ïvö Klack cemented" added "," between "Klack" and "cemented"

Line 688-689: As before, I find this obscure. Why not conclude the sentence by saying 'such
that individuals received the same (high) supply of food.'

A good suggestion, we replaced "but with little competition for space due to the high-energy
environment" with "such that individuals received the same (high) supply of food" to the end of
the sentence.

Line 693: "growth" rephrased to "growth curve"

Line 694: "K –value" deleted the space between "K" and "-value"

Line 699: "more unfavorable or restricting" Not sure if there is much difference (and one
wouldn't say that shallow settings are 'less unfavourable'). Why not substitute 'less favourable'
for both, which would also avoid confusion over what 'former group' refers to?

We thank dr. Johnson for this suggestion and rephrased "more unfavorable or restricting" with
"less favorable"

Line 704: "Von" rephrased to "von"

Line 710: "on" rephrased to "at"

Line 712-714: I think the authors are referring to the spawning which gave rise to each
individual. If so, say 'shows when the individual was spawned' to distinguish from its production
of spawn

We thank dr. Johnson for pointing this out. Indeed, we should have considered the time span
between spawning and settling in our discussion. We revised this part of the discussion to take
this into account. In the light of this and following comments, we rephrased "in which season
spawning occurred" to "in which season the individual settled and started growing its shell". We
edit this in the labels of Figure 11 too.

Line 719: "on" replaced by "at"

Line 719: Broadly, but you need to take into account the larval stage - see comment below. I
suggest you say first growth corresponds to 'time of post-larval settlement'

See above, we rephrase this to "coincides with the time of first growth at post-larval settlement"

Line 721: "(spawning)" rephrased to "(following settlement after the larval stage)"

Line 721: Surely you mean high d18O

Absolutely, we correct this mistake in the revised version.

Line 725: There is only one. Maybe 0.403 needs to be in green because you say 'significantly
influences' at L754. Make title of this column 'Length of growth season'.

We implement these changes in the table

Line 729: "stops" rephrased to "stopped"

Line 730: "are" rephrased to "were"

Line 731: "would be" rephrased to "had been"

Line 733: "should positively correlate" rephrased to "would have positively correlated"

Line 744: Implies that 'temperature of first growth' is not time of spawning - some modifications
needed to take into account the length of the larval stage

Correct, see our reply to previous comments. We take this in account while revising this part of
the discussion.

Line 745: "in" was removed

Line 746: "temperatures" rephrased to "temperature"

Line 756: "extent" rephrased to "range"

Line 762-763: Maybe they correspond to attempted predation

This is an interesting hypothesis and we add it here as a tentative explanation.

Line 765: Oysters can tolerate lower temperatures than this. I think you may mean the lowest
temp. at which growth occurs.

Correct, we rephrase this to "…the lower (~10°C) nor the upper temperature limit of temperature
tolerance (~28°C) between which shell growth occurs…"

Line 767-768: "mild seasonal temperature cycle" rephrased to "mild seasonal temperatures"

Line 773: You have identified the season of spawning as winter in Fig.11b. Actually this is the
time of settlement so you really can't say that spawning was in spring.

Correct, we revise the discussion in this paragraph accordingly

Line 778: It's not clear why. Spring is not a time of max. or min. temperature.

We clarify this by explaining that the input of freshwater dampens the seasonal cycle in $\delta^{18}$O:
"…while also dampening the reconstructed temperature seasonality reconstructed from $\delta^{18}$O
due to the influx of isotopically light fresh water which dampens the seasonal cycle…"

Line 787: "season" replaced by "year"

Line 795-796: OK, but note indirect temperature effects (doi:10.1007/s12237-010-9267-4) and
effects of feeding rate and food type (doi:10.1016/j.gca.2015.07.010) on δ13C

We thank dr. Johnson for the suggested references and added them to this part of the
discussion about carbon isotope ratios in bivalve shells.

Line 799-800: Neither are the low d18O season

Agreed, we made the same mistake as in line 721. This has been rephrased to "high-d18O
season" in the revised text.

Line 800: See comment at L773 re the timing of spawning

Correct, this has been rephrased to "settling of larvae and the onset of shell growth"

Line 812: "and" rephrased to "but"

Line 813: "trends in *R. diluvianum*" rephrased to "trends is inconsistent in *R. diluvianum*"

Line 816: "on" rephrased to "at"

Line 824: "from limited" rephrased to "from the limited"

Line 844: "argue" rephrased to "argues"

Line 849: "disadvantage of the lack of" rephrased to "need for more"

Line 850: "such reconstructions" rephrased to "such proxy-based reconstructions"

Line 855: "However, changes…" rephrased to "However, ontogenetic changes"

Line 875: "with combining" rephrased to "in handling"

**Plaese find our annotated manuscript below:**

[revised manuscript text omitted]